# Bioconjugation strategy for cell surface labelling with gold nanostructures designed for highly localized pH measurement

Leonardo Puppulin[1], Shigekuni Hosogi[1,2], Hongxin Sun[3], Kazuhiko Matsuo[4], Toshio Inui[5,6], Yasuaki Kumamoto [7], Toshinobu Suzaki[8], Hideo Tanaka[7] & Yoshinori Marunaka[1,5,9]

Regulation of intracellular pH is critically important for many cellular functions. The quantification of proton extrusion in different types of cells and physiological conditions is pivotal to fully elucidate the mechanisms of pH homeostasis. Here we show the use of gold nanoparticles (AuNP) to create a high spatial resolution sensor for measuring extracellular pH in proximity of the cell membrane. We test the sensor on HepG2 liver cancer cells and MKN28 gastric cancer cells before and after inhibition of $Na^+/H^+$ exchanger. The gold surface conjugation strategy is conceived with a twofold purpose: i) to anchor the AuNP to the membrane proteins and ii) to quantify the local pH from AuNP using surface enhanced Raman spectroscopy (SERS). The nanometer size of the cell membrane anchored sensor and the use of SERS enable us to visualize highly localized variation of pH induced by $H^+$ extrusion, which is particularly upregulated in cancer cells.

[1] Department of Molecular Cell Physiology, Graduate School of Medical Science, Kyoto Prefectural University of Medicine, Kajii-cho, Kawaramachi-Hirokoji, Kyoto 602-8566, Japan. [2] Department of Clinical and Translational Physiology, Kyoto Pharmaceutical University, 5 Nakauchi-cho, Misasagi, Yamashina-ku, Kyoto 607-8414, Japan. [3] College of Pharmaceutical Sciences, Ritsumeikan University, 1-1-1 Nojihigashi, Kusatsu, Shiga 525-8577, Japan. [4] Department of Anatomy and Developmental Biology, Graduate School of Medical Science, Kyoto Prefectural University of Medicine, Kajii-cho, Kawaramachi-Hirokoji, Kyoto 602-8566, Japan. [5] Research Center for Drug Discovery and Pharmaceutical Development Science, Research Organization of Science and Technology, Ritsumeikan University, Kusatsu 525-8577, Japan. [6] Saisei Mirai Clinics, Moriguchi, 3-34-8 Okubocho, Moriguchi-shi, Osaka 570-0012, Japan. [7] Department of Pathology and Cell Regulation, Graduate School of Medical Science, Kyoto Prefectural University of Medicine, Kajii-cho, Kawaramachi-Hirokoji, Kyoto 602-8566, Japan. [8] Department of Biology, Graduate School of Science, Kobe University, 1-1 Rokkodai-cho, Nada-ku, Kobe 657-8501, Japan. [9] Research Institute for Clinical Physiology, Kyoto Industrial Health Association, 67 Kitatsuboi-cho, Nishino-kyo, Nakagyo-ku, Kyoto 604-8472, Japan. Correspondence and requests for materials should be addressed to L.P. (email: leonardo@koto.kpu-m.ac.jp) or to Y.M. (email: marunaka@koto.kpu-m.ac.jp)

 

The intracellular pH in most living cells is alkaline and cell life is possible only if variations of proton concentration are kept within a very narrow range[1,2]. In addition to buffering systems acting in the cytosol, such as the bicarbonate system and phosphoric acid, several membrane transporters are responsible for maintaining the correct pH in the cytosol by extruding protons against the electrochemical potential gradient and they play primary roles in maintaining alkaline pH inside cells[3–5]. For example, in renal tubular cells the sodium hydrogen exchanger (NHE), the sodium-dependent and -independent chloride-bicarbonate exchanger ($Cl^−–HCO_3^−$), the sodium bicarbonate co-transport ($Na^+–HCO_3^−$), the ATP-dependent proton pump ($H^+–ATPase$), and the ATP-dependent proton–potassium pump ($H^+–K^+–ATPase$) regulate pH home-ostasis[6]. Abnormal intracellular pH, which can be caused by impairment of these transporters, is associated with dysfunction of cells, diseases, and decrease in physical performance. In addition, as far as the study of cancer cells is concerned, it has been demonstrated that cellular pH is crucial for biological functions such as cell proliferation, metastasis, drug resistance, and apoptosis[7,8]. Acidification of the extracellular milieu is expected in cancer tissues, mainly due to elevated cell glycolytic activity[7, 8] (i.e., Warburg effect) that upregulates proton extrusion to maintain the intracellular pH within a physiological range. Although interstitial pH reduction can be detected using confocal fluorescence microscopy (CFM), no experimental techniques have been heretofore available for visualizing highly localized upregulation of $H^+$ membrane transporters. In fact, for this purpose, the pH sensor is required to be of nanometer size and located at the point of proton extrusion. In most of the studies exploiting confocal fluorescence imaging, however, the pH-sensitive probing molecules were dissolved in the intracellular and extracellular compartments, namely the reported values represent the average pH inside the micrometric laser probe[9–11]. An interesting new approach has been recently proposed based on the design of a low-pH insertion peptide conjugated to a pH-responsive fluorescent dye, but this method is limited to the study of cancer cells in which the interstitial pH in proximity of the membrane is sufficiently acidic to enable the peptide insertion[12]. Magnetic resonance spectroscopy (MRS) is another alternative noninvasive experimental technique exploited to measure extra-cellular pH using endogenous or exogenous pH-sensitive molecules[13–16]. Although a more sophisticated but cumbersome approach exploiting magnetic resonance force microscopy is reported to reach spatial resolution of 90 nm[17], conventional MRS possesses spatial resolution ranging from millimeters to micrometers and it cannot measure pH on a single cell level[12,17]. Functionalization of gold quasi three-dimensional plasmonic nanostructure array with 4-mercaptobenzoic acid (4-MBA) has been recently proposed as a clever experimental approach to measure extracellular pH in proximity of the basal outer mem-brane of cells[18]. In this method a uniform self-assembled monolayer (SAM) of 4-MBA was conjugated to the plasmonic nanostructured substrate upon which cells were seeded. Surface enhanced Raman spectroscopy (SERS) was then exploited to measure the pH-dependent concentration of deprotonated 4-MBA. Although this approach enabled reproducible mapping of extracellular pH, the level of acidification measured on HepG2 human liver cancer cells was much inferior than the typical acidosis expected on the surface of cancer cells[12,19–21], which indicates that the pH probe on the substrate may not have been entirely in contact with the cell surface. Gold nanoparticles (AuNP) conjugated with 4-MBA were also previously exploited to measure pH in cells, but only after nonspecific endocytosis of the nanosensors, namely in endosomes and lysosomes[22–26]. Based on these preliminary considerations, the development of new advanced methods of analysis is of pivotal importance to gain a deeper understanding of the pH regulation mechanisms in different types of cells. In the attempt of filling this gap of knowledge, we exploit here the remarkable optical properties of AuNP and their ability to conjugate with different thiol-containing molecular compounds to develop a method for highly localized pH bio-sensing using SERS. The strategy for AuNP conjugation is specifically designed to efficiently target the cell membrane proteins and to quantify the local pH by collecting the Raman scattering of the 4-MBA monolayer assembled on the gold surface. Experiments on HepG2 human liver cancer cells and MKN28 gastric cancer cells prove the successful anchoring of the AuNP to the outer membrane and show substantial acidification of the extracellular surface pH. We also detect the clear increase of cell surface pH after addition of ethyl-isopropyl amiloride (EIPA), an inhibitor of NHE, proving the sensitivity of the sensor to dynamic variations of proton trafficking.

## Results

**Outer cell membrane surface labelling.** An explanatory sketch of the AuNP-based pH nanosensor is shown in Fig. 1a. The strategy devised to label the membrane proteins of cells followed a multistep approach. First, we labelled the outer membrane surface of the cells using a sulfo-NHS-ester-biotin compound (NHS-B) that reacted with the primary amines of lysine and the amino-termini of polypeptides. In a separate step, we added 4-MBA and a pyridyldithiol-biotin compound (HPDP-B) to the AuNP colloidal solution, which conjugated via thiol-gold interaction. Lastly, streptavidin (SA) provided the strong and selective bond between conjugated AuNP and biotinylated membrane surface proteins. Differently from previous studies that pioneered the cell surface labelling with nanostructures[27,28], in our protocol SA first reacted with biotinylated surface proteins; then, only after removal of unreacted SA, we added the conjugated AuNP to the cell sample. This is a crucial feature of the experimental method, since it guaranteed that each AuNP attached to the cell membrane had approximately only one SA molecule close to its surface, namely the SERS signal of SA did not affect the overriding signal of the pH-sensitive 4-MBA. We also avoided simultaneous conjugation of AuNP with HPDP-B and SA to reduce the probability of micro-aggregation due to the possible interaction of SA with HPDP-B molecules reacted with two different AuNP. Similar to this experimental approach, by utilizing specific antibodies instead of NHS-B, it is possible to engineer different strategies tailored to the study of specific membrane receptors[27]. We developed our protocol for cell surface labelling by optimizing the yield of each reaction involving the selected biochemical compounds. The validity and applicability of this method were tested in gastric and lever cancer cells. We carried out diversified experiments by CFM, SERS, and transmission electron microscopy (TEM) to present strong evidence supporting the success of AuNP attachment to the external membrane without endocytosis during the time of pH measurement. As a first step in the development of this method, we investigated the AuNP distribution on the membrane by monitoring the intensity of Alexa Fluor® 488 fluorescence dye covalently linked to streptavidin (hereafter referred as Alexa-SA), which bound to the AuNP through the biotin moieties of HPDP-B, as shown in the simplified sketch of Fig. 1b. We collected confocal $xy$ raster scans of fluorescence intensity at different focal positions along the $z$-axis, from the glass bottom ($z = 0$) up to above the cells with 1 μm step. The CFM experiments were carried out more than 1 h after the completion of the protocol for AuNP attachment. Figure 2b–f shows the fluorescence images collected from the cell cluster of Fig. 2a at $z = 0, 6, 12, 14,$ and 16 μm, respectively. Hoechst nucleic

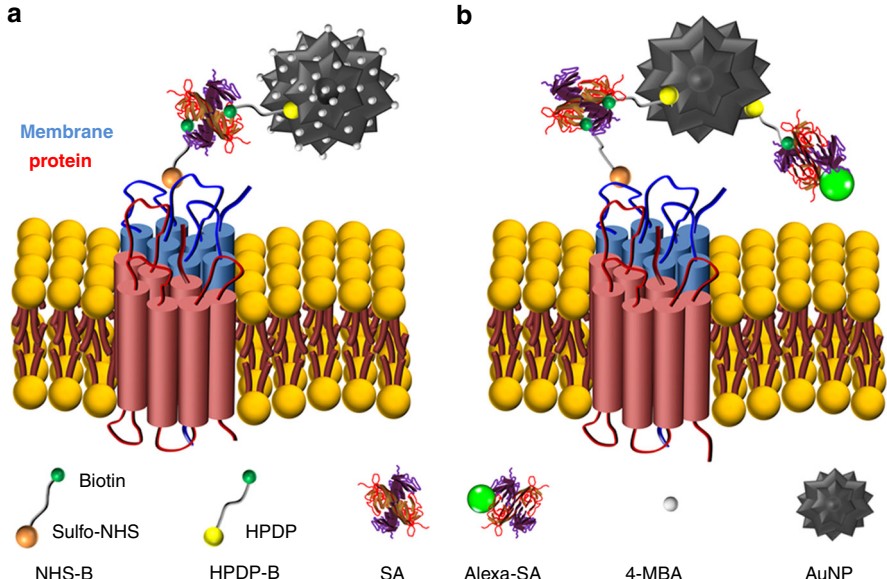

**Fig. 1** Bioconjugation strategy for cell surface labelling using AuNP. **a** Explanatory sketch illustrating the structure of the pH nanosensor: the surface of AuNP is functionalized with 4-MBA and HPDP-B. The sulfo-NHS moiety of the biotinylation reagent NHS-B reacts with primary amines in lysines or amino-termini of membrane proteins such as receptors, pores, channels, carriers, pumps, integrins, or enzymes. Two of the four active sites of streptavidin (SA) provide the link between the biotins of NHS-B anchored to proteins and HPDP-B conjugated to AuNP. **b** Conjugated AuNP labelled with Alexa-SA: the attachment of the nanoparticles to the cell membrane was confirmed using AuNP functionalized with HPDP-B, some biotins of which reacted with Alexa-SA to enable visualization by CFM

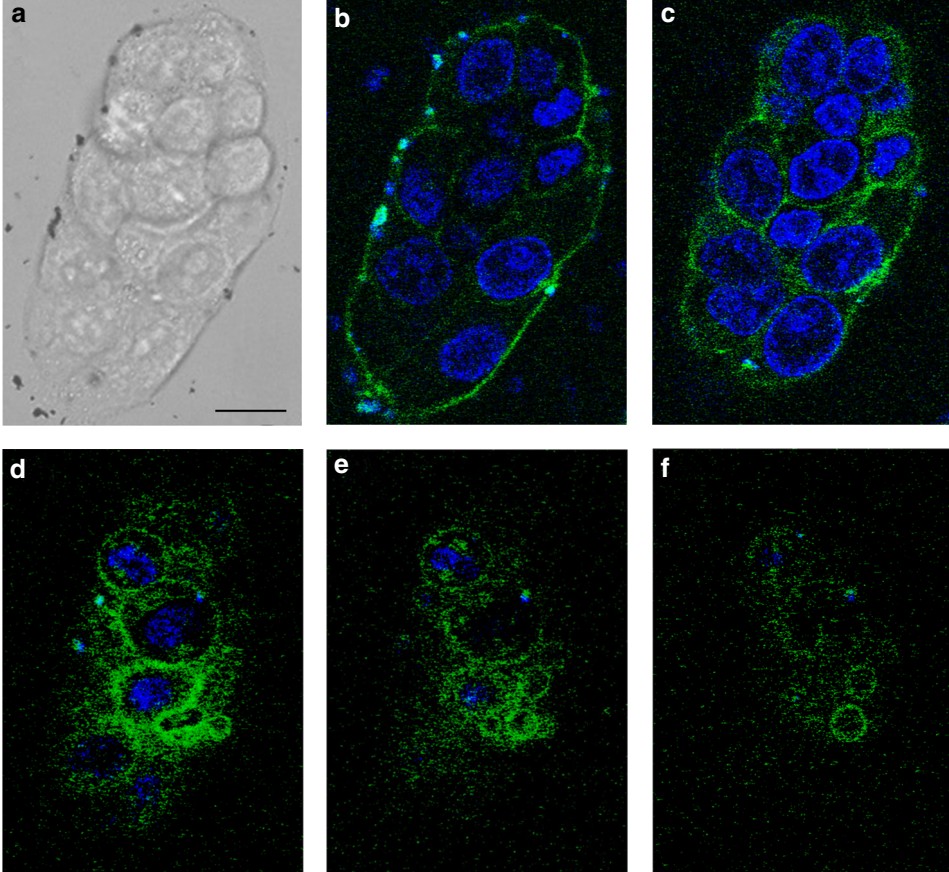

**Fig. 2** Gold nanoparticles distribution on the cell outer membrane surface detected by CFM. The analysis was performed on the cluster of MKN28 cells shown in **a** and treated with the AuNP of Fig. 1b according to our protocol. In **b**–**f** are reported the fluorescence images collected at $z = 0, 6, 12, 14$, and $16$ μm, respectively. Green is the Alexa488 dye linked to the AuNP, while the nuclei were stained with blue Hoechst 33342 dye. In some locations of the glass bottom dish, close to the cell cluster at $z = 0$, we noticed microscopic AuNP aggregations, in which traces of Hoechst dye were detected. In Supplementary Movie 1, we combined the collected images to create a movie of the $z$-stack sections through the cells. Scale bar: 10 μm

acid fluorescent stain was used to identify the cell nuclei, which occupy the most of the cytosolic space in cancer cells. Although some micro-aggregations were formed at the edge of the cell cluster, these results demonstrated the widespread presence of the nanosensor on the cell surface and the absence of incorporation in the cytosol and the nuclei. In addition, we investigated the reactions of surface protein biotinylation by linking Alexa-SA to the biotin moieties of NHS-B reacted with surface proteins, as described in Supplementary Fig. 1a. This experimental procedure was conceived to optimize the NHS-B reaction and distribution on the membrane surface in our biotinylation protocol. Supplementary Fig. 1c–g shows the merged fluorescence images of nuclei and NHS-B collected from the cell cluster of Supplementary Fig. 1b at $z = 3, 6, 10, 13$, and 18 μm from the bottom glass, respectively. From these outcomes, it is clear that we obtained uniform distribution of NHS-B on the cell surface and the sulfo group in the molecular label prevented internalization, as expected. We optimized this step of the experimental protocol by repeating the biotinylation at different pH of the buffer solution: more alkaline pH = 8 showed the highest peaks of NHS-B fluorescence, as shown in Supplementary Fig. 2. In fact, the yield of aminolysis between primary amine and the ester group of NHS-B is reduced in an acidic environment (i.e., pH < 7)[29], as expected in the proximity of the cancer cell outer membrane. The use of PBS buffer at pH = 8 during bitotinylation maximized the presence of NHS-B on the surface without inducing cell death during the 30 min treatment. Confocal fluorescence microscopy gave compelling evidence about the dependability of the strategy for cell surface labelling and it contributed to the calibration of the protocol in terms of optimal buffer solution pH and concentration of the molecular compounds for conjugation. Nonetheless, the conclusive experimental confirmation of AuNP surface anchoring was provided by TEM. Figure 3a–e shows examples of TEM images at different magnifications collected from 90 nm thick sections of MKN28 cells embedded into resin after surface treatment, rinsing, 1 h waiting and fixation. The waiting time before cell fixation was estimated as the typical time spent for SERS experiments, in order to confirm the lack of endocytosis during pH assessment. According to TEM images, both isolated AuNP and nano-aggregations were anchored to the outer surface and the microvilli, which are an integral part of the cell membrane in MKN28 cells. Although proteins of the plasma membrane are continuously internalized causing AuNP endocytosis, their specific turnover rates are ranging from hours to several weeks, which explain the lack of AuNP in the cytosol during the time frame of our experiments[30]. Moreover, Cheng et al.[28] showed lack of endocytosis in their nanoparticles after attachment to the cell surface for up to 2 days. They suggested that the cause may be the aggregation of nanoparticles binding with multiple membrane proteins. The in-plane spot size of the laser probe was estimated as 700 nm (see details of the experimental procedure in section Methods, SERS analysis). In Fig. 3b, we report an explanatory description of the SERS pH-probe given by the interaction between the laser beam and the AuNP. The in-plane spatial resolution of the measurement can be approximated as the size of the laser spot. Conversely, we define the axial resolution as the maximum distance from the cell surface at which the pH is measured. Since the SERS signal is merely originated from the 4-MBA SAM (i.e., 0.78 nm thickness[31]), depicted with green lines in Fig. 3b, the axial resolution can be approximated as the size of the nanosensors. According to TEM images, it can be estimated as ranging from 90 nm, i.e., single AuNP or one-dimensional aggregation lying on the membrane surface as in Fig. 3e to 400 nm (i.e., the largest cluster of AuNP observed in the TEM images). The previous experimental evidences proved the success of the AuNP attachment to the cell

outer surface and confirmed the highly localized nature of this method for pH assessment. As a final validation step of the protocol, we tested cell viability by MTT assay. The absorbance of formazan to quantify the number of living cells was measured 3 h after the completion of the protocol and the results are shown in Fig. 4a. Cell viability was expressed as percentage of living cells with respect to the average of the control samples (i.e., $n = 3$ independent experiments for treated and untreated cells). As compared to their controls, MKN28 and HepG2 labelled with AuNP did not show statistically significant difference in cell viability (two-tailed unpaired $t$ test, $n = 3$, $p = 0.803$ and $0.533$, respectively). In addition, fluorescent propidium iodide (PI) staining provided further evidence regarding the lack of cell apoptosis induced by the treatment. Figure 4 reports bright-field (b), Hoechst nuclei dye fluorescence (c), PI dye fluorescence (d), and merged fluorescence (e) images from a location of MKN28 control sample. Similarly, Fig. 4f–i shows the images collected from a sample of AuNP treated MKN28 cells. Hoechst blue-fluorescent dye can bind to DNA of living and dead cells. PI is a red-fluorescent dye binding to DNA, but it is permeant only to dead cells. The fluorescence microscopy images were collected at least 1 h after the completion of AuNP treatment. The number of dead cells highlighted by PI was similar in the two investigated cases and it can be considered negligible as compared to the total number of cells. The region inside the dotted white circles in Fig. 4f–i was selected to point out the occurrence of nuclei fragmentation that was not ascribable to cell apoptosis, but, conversely, to cell mitosis, which represents an additional proof of the normal physiological activity of the cells.

**Measurement of pH using AuNP and SERS.** One of the most interesting effects of the interaction between molecules and the electromagnetic field of a monochromatic laser is the generation of molecular vibrations that alter the frequency of the scattered light, which is known as inelastic or Raman scattering. The Raman light intensity is proportional to the concentration of moieties from which the vibrations originate. Unfortunately, conventional methods of analysis based on Raman spectroscopy are not feasible in physiology to visualize molecular concentrations in the micro/nanomolar range, since the Raman scattering is weak and undetectable at those low molarities. Gold nanoparticles can localize plasmon polaritons in their close vicinities once illuminated with a laser beam of a suitable wavelength in resonance with plasmons. The plasmon polaritons can generate a giant electromagnetic field at the metal surface. SERS can exploit this physical phenomenon to detect the Raman scattering of molecules located on the AuNP surface even at concentrations as low as picomoles[32]. Over the past two decades, 4-MBA SAM on metal nanosubstrates have been intensely explored because this molecule can readily conjugate to gold substrates and its surface plasmon enhanced Raman scattering is strong and depends on the pH of the surrounding nanoenvironment. Such a peculiar combination of advantages makes this molecule an attractive choice for the development and production of pH nanosensors. Figure 5a shows a representative SERS spectrum collected from 4-MBA/HPDP-B (concentration ratio 1000:1) conjugated AuNP colloidal solution at pH 6.4 in the spectral range 950–1750 cm$^{-1}$. Band fitting was performed using Gaussian-Lorentzian (i.e., Voigtian) functions after baseline subtraction. The intensity detected at around 1400 cm$^{-1}$ belongs to the COO$^-$ symmetric stretching, namely it merely originates from deprotonated 4-MBA molecules and it can be used to quantify pH, as already demonstrated in several previous studies[18,22–26,33–35]. The majority of the investigations on 4-MBA SAM quantified the total intensity of the COO$^-$ symmetric stretching by considering only

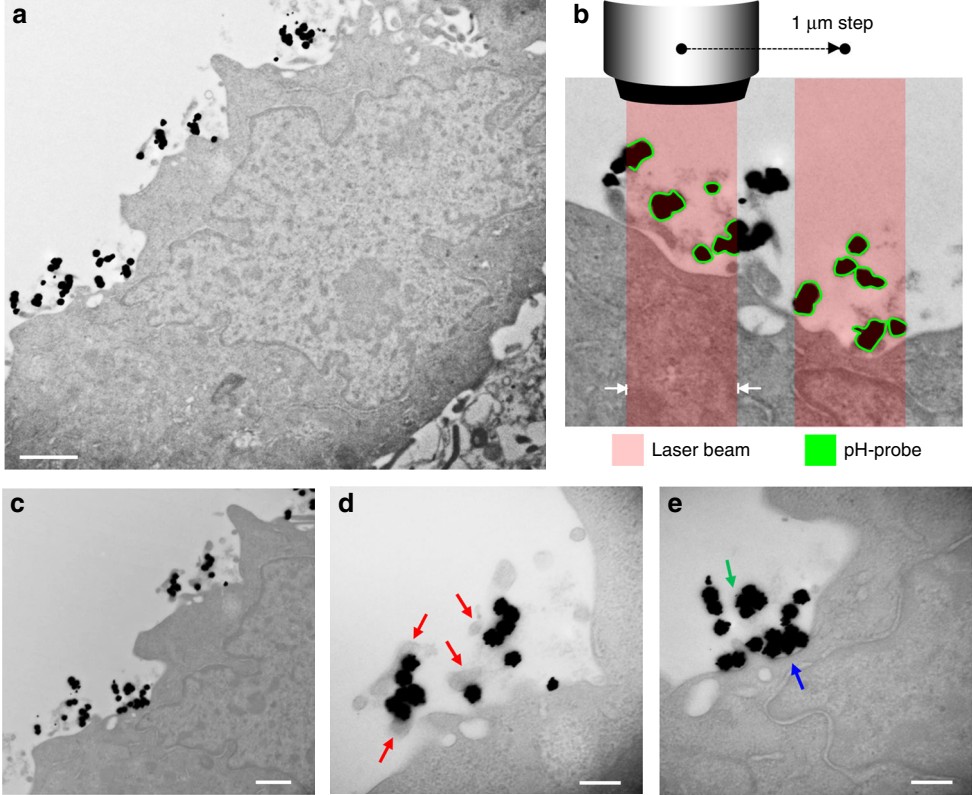

**Fig. 3** Confirmation of AuNP anchoring to the outer cell surface by TEM. Images were collected from MKN28 cells. The cells were fixed after 1 h from the completion of AuNP attachment, in order to confirm also the lack of endocytosis during SERS experiments. The cells embedded into resin were trimmed through their thickness using an ultramicrotome to obtain 90 nm thin sections. The images in **a** and **b** were taken from the same section, but at different magnification (7 and 17 k, respectively). Similarly, images in **c**–**e** belong to another section of the same cell at magnification 17, 50, and 50 k, respectively. In **b** the pH-sensitive 4-MBA SAM of each AuNP illuminated by the laser beam is depicted with green lines. The thickness of the lines does not represent the real thickness of the SAM, which can be assumed as 0.78 nm[31]. Red arrows in **d** indicate sections of microvilli. The 90 nm thickness of the slices for TEM analysis is equal to the average size of the AuNP, namely some of the AuNP were trimmed by the diamond knife. For this reason, in the nanosensors indicated by the green arrow in **e** the point of contact with the microvillus may have been removed during trimming. The blue arrow in **e** points out an aggregation of AuNP lying on outer cell membrane, whose maximum thickness can be estimated as about 100 nm. Scale bars: 1 μm in **a**, 500 nm in **c**, 200 nm in **d** and **e**. In **b** the laser beam waist is 700 nm (length between the two white arrows)

one band. In the numerical fitting routine, the asymmetric nature of this band at neutral and alkaline pH was neglected and the observed red shift as the solution becomes more acidic was attributed to hydrogen bonding between some adjacent 4-MBA molecules and the interaction between carboxylate groups and ring hydrogens[18,22,35]. In the present investigation, we deconvoluted the intensity originated from the $COO^-$ symmetric stretching into two sub-bands centered at 1390 and 1415 $cm^{-1}$, which were assigned to carboxylate groups subjected to intra-monolayer bonding and unbonded carboxylates of more vertically oriented 4-MBA molecules, respectively. Systematic fitting procedure based on this band interpretation enabled us to obtain the most consistent results in term of pH estimation. Among the other spectral features of 4-MBA, the prominent bands located at 1068 and 1588 $cm^{-1}$ are due to $\nu_{12}$ and $\nu_{8a}$ aromatic ring vibrations, respectively, while the band at 1710 $cm^{-1}$ is assigned to the C = O stretching of the COOH group (i.e., protonated 4-MBA molecule). These five bands were labelled in Fig. 5a as band A–E, respectively. In Fig. 5b, we reported the SERS spectrum of HPDP-B in AuNP colloidal solution (50 μM), the same concentration of 4-MBA in Fig. 5a. The HPDP-B molecule is composed of a pyridyldithiol group linked to a biotin residue via a 2.9 nm hexyl arm spacer. The disulfide bridge reacts with the gold surface[36,37], leading to dissociative adsorption of HPDP-B through gold-adatom-mediated bonds as 2-pyridine (2-Py)

thiolate and biotin-hexyl spacer arm thiolate. Compared to the previously reported SERS spectrum of HPDP-B on silver nanorod substrates[38], we observed the most intense bands at 1001, 1051, 1081, and 1546 $cm^{-1}$, ascribable to ring vibrations of 2-Py thiolate[39,40]. The SERS intensities collected at 1220, 1370, 1448, 1466, and 1609 $cm^{-1}$ were assigned to molecular vibrations of biotin and hexyl arm spacer[41,42]. Supplementary Table 1 and Supplementary Fig. 3 report the band assignments[39–43] and the comparison between experimental spectra of HPDP-B and 2-Py thiolate obtained from the Raman analysis of AuNP colloidal solutions after conjugation. The most intense bands of 2-Py thiolate can be clearly observed also in the HPDP-B spectrum. Considering that the concentration ratio 4-MBA:HPDP-B selected for the assembly of the pH-sensor is 1000:1, the Raman scattering cross-section of HPDP-B is also ~3 order of magnitude smaller than the cross-section of 4-MBA, which explains the absence of HPDP-B Raman bands in the spectrum of Fig. 5a. Figure 5c shows the spectrum obtained from AuNP conjugated with thiophenol (TP): 4 intense bands were detected at 999, 1021, 1063, and 1578 $cm^{-1}$, which correspond to the locations of bands observed also in the spectrum of Fig. 5a and labelled as F–I. Such experimental evidence suggests that catalytic decarboxylation of 4-MBA may have occurred on the surface of AuNP exposed to laser plasmon enhancement, leading to formation of TP[33,44]. This reaction was favored at alkaline pH (i.e., pH > 9 in our

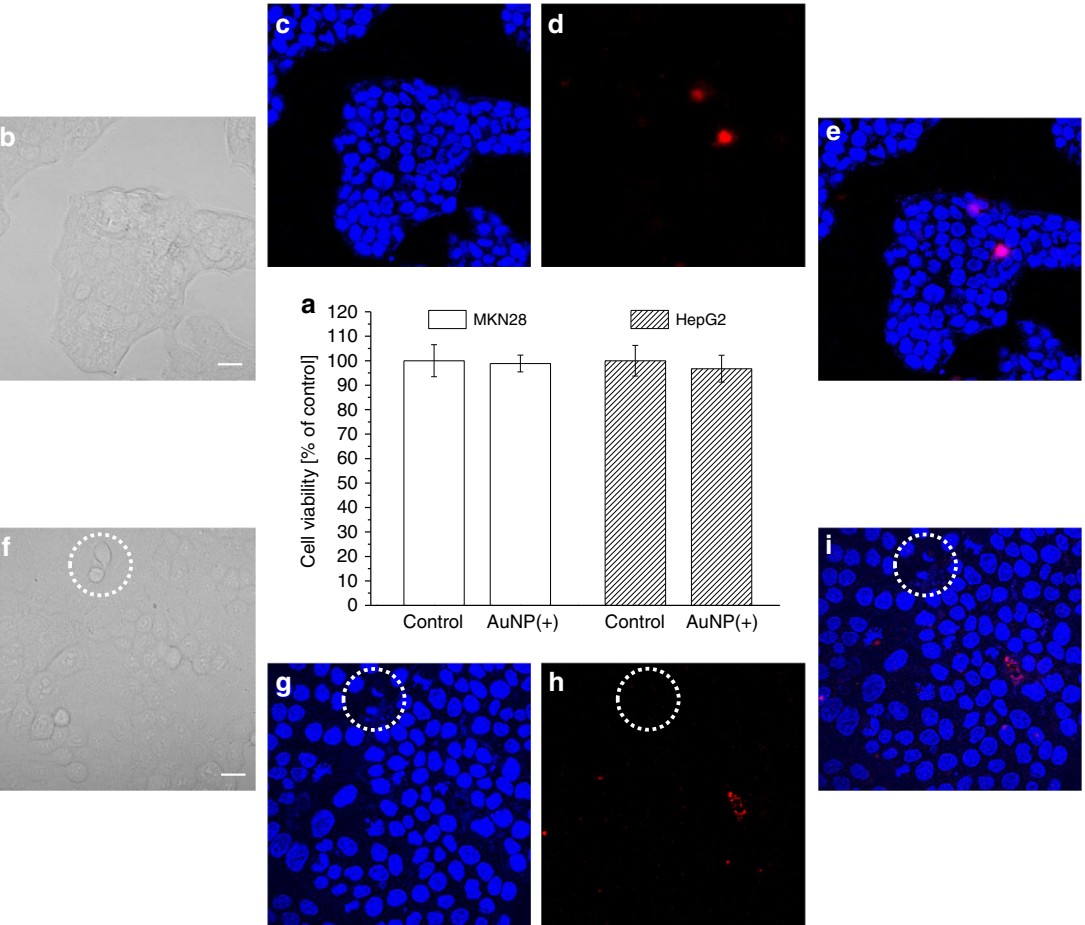

**Fig. 4** Cytotoxicity assay after cell surface labelling with pH-sensitive AuNP. **a** MTT assay results reported as percentage of living cells with respect to the average of the control samples. Both MKN28 and HepG2 labelled with AuNP do not show statistically significant difference with respect to their controls (two-tailed unpaired $t$ test, $p < 0.05$, $n = 3$ independent experiments, error bars show the standard deviation from the mean). **b** Bright-field, **c** Hoechst nuclei dye fluorescence, **d** PI dye fluorescence, and **e** merged fluorescence images collected from the same location of a control sample of MKN28. Similarly, **f–i** report the images collected from an AuNP treated MKN28 sample, respectively. The region inside the dotted white circle show the occurrence of nuclei fragmentation during cell mitosis, which represents an additional proof of normal physiological activity of the cells. Scale bars: 20 μm. Source data are provided as a Source Data file. Source dataSource data

experiments), where the population of deprotonated 4-MBA increased along with the probability of plasmon-induced catalytic reaction forming $CO_2$ and TP[44,45]. Confirmation of the aforementioned modification of the 4-MBA SAM was provided by the results presented in Fig. 6. We collected series of SERS spectra of 4-MBA from AuNP colloidal solution (pH = 7.4) at different power outputs of the laser. In Fig. 6a we compared 3 typical spectra obtained at 1, 20, and 72 mW, which were normalized to the peak intensity detected at around 1068 $cm^{-1}$. For clarity sake, in Fig. 6b–d we plotted the experimental intensities and the band fitting in the range 980–1120 $cm^{-1}$ for the spectra acquired at 1, 20, and 74 mW, respectively. Similarly, also the intensities in the range 1350–1450 $cm^{-1}$ are shown in Fig. 6e–g. According to these results, it follows that the onset of bands F–H was strongly favored by the increase of the power. At 1 mW the energy concentrated on the surface hot spots was not sufficient to trigger modification of the SAM. Moreover, from 20 to 72 mW, the increase of the intensities of F–H bands was accompanied by the reduction of the ratio between bands B and A, namely the unbonded $COO^-$ group was more prone to dissociate. At the medium power of 20 mW the transformation was not observed in every collected spectrum, probably because during the 20 s acquisition time on colloidal solution the Brownian motion of AuNP affected the probability of transformation in some

measurements. Sun et al.[46] reported similar variations on the SERS spectra of 4-mercaptophenylboronic acid (4-MPBA) SAM reacting with fructose. They hypothesized that the origin of these bands may be due to reorientation and charge redistribution of the benzene ring. Figure 6h shows that normalizing the symmetric stretching of deprotonated $COO^-$ group (i.e., band A + B) with the intensity of band C, the ratio was independent on the laser power, while normalization by using the total intensity of band C + H lead to great differences. It is clear that, when F and G bands were observed in the 4-MBA SERS spectrum, deconvolution of bands C and H was necessary for the correct chemical–physical characterization of the SAM in our nanosensor. These arguments can be considered as a preparatory step to establish the criteria for pH calculation. In fact, in all the studies published so far, the pH-dependent intensity of the $COO^-$ symmetric stretching (i.e., band A + B) was normalized using the intensity of $\nu_{12}$ or $\nu_{8a}$ aromatic ring vibrations, even in presence of the bands F–I. Differently, following the spectral fitting routine previously explained, we considered the ratio $R$ of band A + B to band C (or D), neglecting the deconvoluted intensity of band H (or I), if detectable. The results reported hereafter were obtained using band C, that was preferred to band D due to the slightly lower standard deviation of $R$ calculated from experiments on solutions with the same pH. The plot of $R$ as a function of pH is

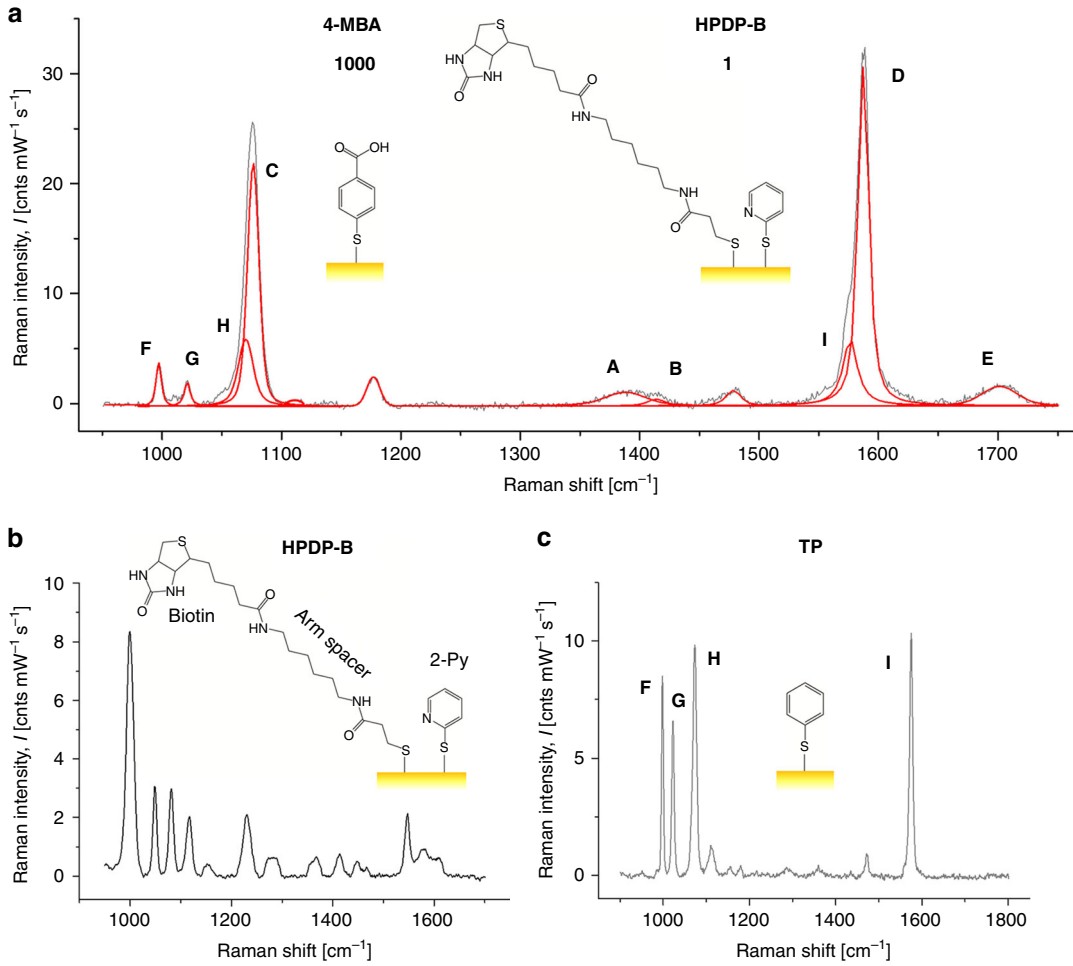

**Fig. 5** Interpretation of the SERS intensity bands collected from the pH nanosensor. Representative SERS spectra of **a** 4-MBA/HPDP-B (ratio 1000/1), **b** HPDP-B, and **c** thiophenol (TP) conjugated AuNP. In **b** and **c** the HPDP-B and TP concentrations were equal to the concentration of 4-MBA used in **a**. In **a** the bands of the spectrum were fitted using Voigtian functions and the assignments of 4-MBA and TP vibration modes, which enabled mathematical deconvolution of the intensities of bands A + B, C + H and D + I

shown in Fig. 7a and it follows the typical trend of acid–base dissociation described by the Henderson–Hasselbalch equation, which was used to fit the data and to obtain the following phenomenological correlation:

$$\text{pH} = 6.99 + \ln\frac{R - 0.025}{0.490 - R}. \tag{1}$$

We used Eq. (1) to quantify the extracellular pH on the membrane of cells treated with functionalized AuNP. The experimental data for the calibration curve were obtained from the buffer solution used for cell analysis adjusted to different values of pH in the range 4–11, in which we dispersed the conjugated AuNP after separation by centrifugation ($n = 8$ measurements for each value of pH). Figure 7b, c report examples of spectra collected at pH 6.0 and 7.4, in which was clear the intensity variation of bands A, B, and E depending on the proton concentration in solution. For clarity sake, we purposely selected two spectra with comparable influence of F–I bands. In addition, we calculated the ratio R using the conventional approach, namely normalizing the band A + B to band C + H. In this case, the mean of the standard deviations calculated from solutions at different pH is 0.052, while for the data showed in Fig. 7a, the average standard deviation is 0.016. Such a difference in standard deviation of R reflects the error introduced by considering the intensity of

decomposed aromatic species (i.e., TP) as a normalizing factor. The final characterization of the functionalized AuNP was the estimation of the SERS enhancement factor (EF), which was equal to $1.67 \times 10^6$, according to the procedure reported in the section Methods.

**Hyperspectral pH sensing on the outer membrane of cells.** Using SERS, we analyzed MKN28 and HepG2 cells after anchoring of conjugated AuNP in accordance to the protocol validated in the previous sections. We made preliminary acquisition of Raman spectra through 20 μm z-axis line scans in the middle of the cells, from the glass bottom up to above the apical surface. As shown in Fig. 8, the SERS maximum intensities of each spectral band were localized in a single z-position, which represents the location of the AuNP attached to the cell surface. Supplementary Fig. 4a, b reports the profiles of fluorescence and SERS intensities obtained from z-axis line scans through one 50 nm fluorescent bead attached to the glass substrate and the AuNP of Fig. 8, respectively. The experimental trends were fitted using the Lorentzian function describing the intensity axial profile of the laser probe. According to the full width at half maximum (FWHM) calculated from the experiment on one fluorescent bead, the axial resolution of the laser probe was estimated equal to 3.1 μm. The Lorentzian trend in Supplementary Fig. 4b from the

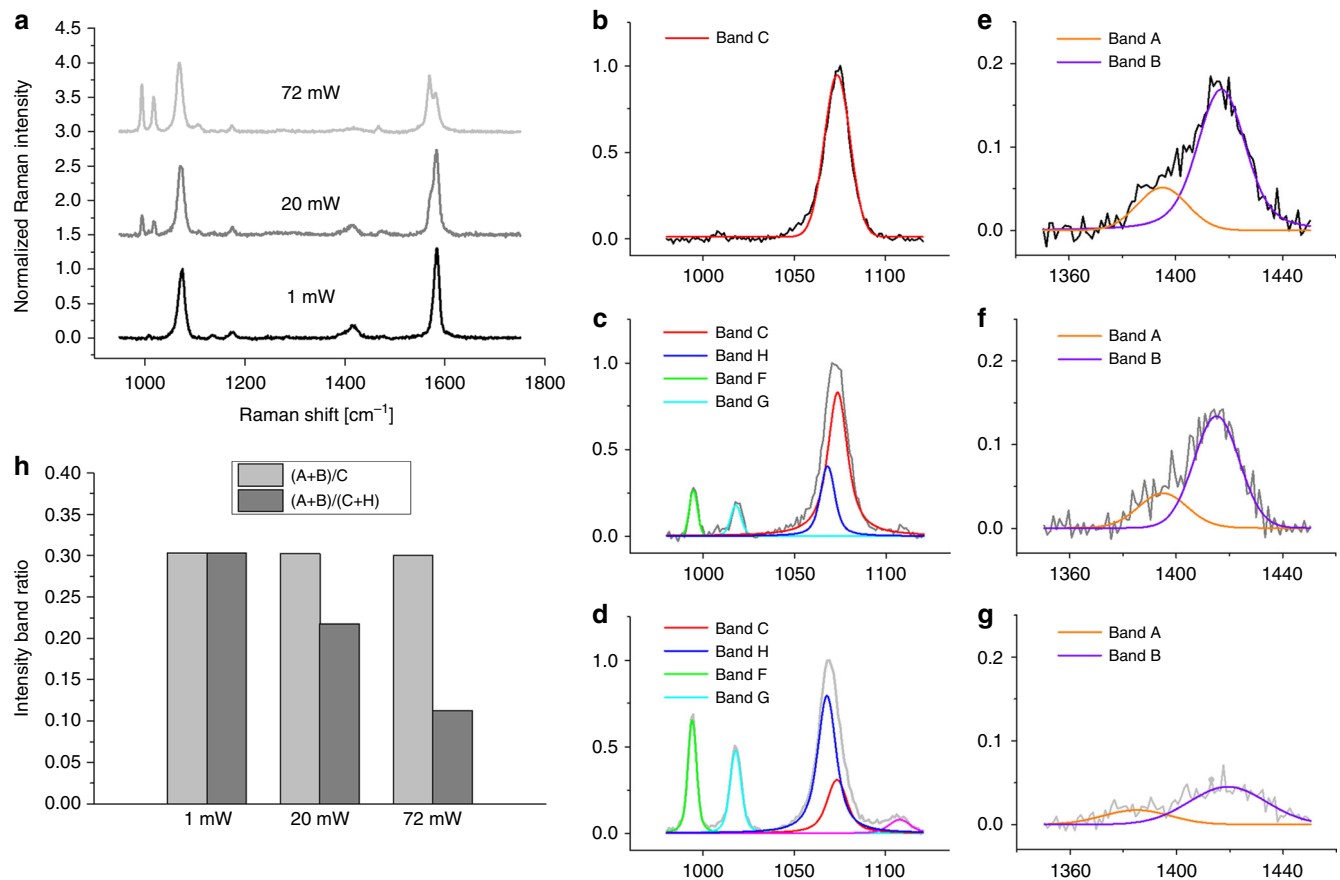

**Fig. 6** Modification of the 4-MBA SAM during SERS analysis. **a** Typical 4-MBA spectra obtained from the same AuNP colloidal solution (pH = 7.4) at 1, 20, and 72 mW, normalized to the intensity detected at around 1068 cm$^{-1}$. **b–d** Experimental intensities and band fitting in the range 980–1120 cm$^{-1}$ from the spectra in **a** acquired at 1, 20, and 74 mW, respectively. In **e–g**, intensities and band fitting results are shown in the range 1350–1450 cm$^{-1}$ as acquired at 1, 20, and 74 mW, respectively. **h** Comparison of the intensity band ratios (A + B)/C and (A + B)/(C + H) calculated at different outputs of the laser source, from the spectra shown in **a**

AuNP attached to the cell surface gave comparable FWHM of 3.8 μm. Two dimensional patterns of pH distribution were obtained collecting SERS spectra from the AuNP and nanoaggregates anchored to the cell surface. Figure 9a–d reports hyperspectral maps of pH collected from MKN28 and HepG2 cells in different physiological conditions. The z-coordinate of each x–y raster scan was set after estimation of the average AuNP position in the middle of each analyzed cell by z-axis line scans. Considering the micrometric axial length of the laser probe, this procedure enabled to illuminate the AuNP attached to the surface in the majority of the investigated spots. The pH was not measured at the edges of the cells, since in those locations the cell thickness steeply decreased and the AuNP were out-of-focus. For each xy raster scan, we also show the Raman image of the band C intensity, which highlights the poor signal detected also in some locations on the cell surface, in which we could not reliably calculate the values of pH. The presence of micro-aggregations was noticed on the edge of some cells. In Supplementary Fig. 5, we present an example of spectrum collected from microscopic aggregations, which was much more intense than the typical spectra obtained on the cell surfaces. The average pH calculated from spectra of aggregations was 7.41, which was almost equal to the pH of the buffer solution. The clear separation between the intensities detectable on micrometer and nanometer scale AuNP clusters was demonstrated by the results shown in Fig. 9e. The spectra considered for the assessment of cell surface pH in this study were comparable to those measured from colloidal

solutions, while they were typically two orders of magnitude less intense than spectra referred to the micro-aggregations. Figure 9a, b shows two hyperspectral maps of pH collected from MKN28 and HepG2 cells in pH 7.4 buffer solution, respectively. According to our findings, the mean surface interstitial pH in MKN28 cells was 6.2 ± 0.2 (n = 3 cells); the most acidic and alkaline values were estimated to 5.0 and 7.4, respectively. The mean standard deviation of pH in a single cell was 0.4, which denotes highly localized variations of proton concentration on the outer surface of the cell. In the case of HepG2 cells, the average pH was 6.7 ± 0.1 (n = 3 cells), while the most acidic and the most alkaline values were 5.1 and 7.5, respectively. The increase of [H$^+$] detected in our experiments was presumably correlated to upre-gulation of V-type H$^+$–ATPase, NHE, monocarboxylate trans-porters (MCTs), and carbonic anhydrases (CAs), which is a peculiarity of most cancer histotypes[47–49]. In order to confirm this hypothesis and to test the sensitivity of this method for measurement of surface pH, we also carried out experiments on MKN28 cells treated with EIPA, an inhibitor of NHE, and on HepG2 cells after fixation by glutaraldehyde/paraformaldehyde. Figure 9c shows a representative pH map performed 30 min after the application of EIPA in MKN28 cells, namely during the acute phase of NHE blockade. The mean surface interstitial pH on EIPA-treated MKN28 increased to 6.7 ± 0.04 (n = 3 cells). In some locations we still detected pH lower than 6.5, which may be correlated to the higher activity of V-type H$^+$–ATPase and MCTs. Measurements of cytosolic pH in MKN28 cells as a

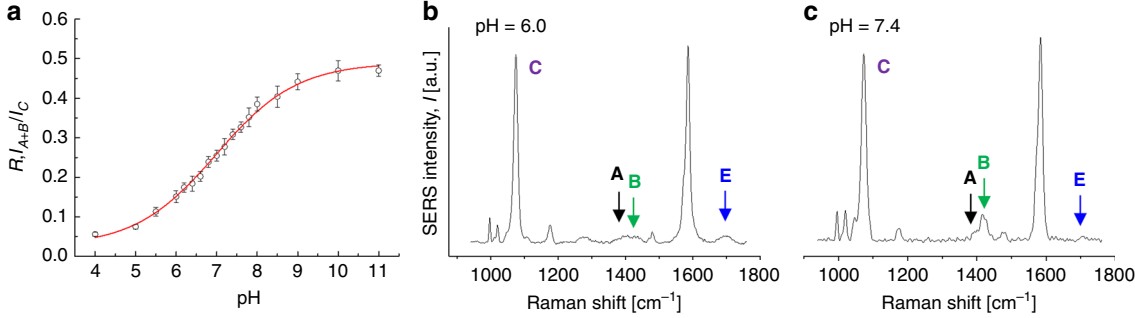

**Fig. 7** SERS intensity response to the variation of pH. **a** Experimental calibration curve of ratio $R = I_{A+B}/I_C$ as a function of pH (error bars show the standard deviation from the mean of $n = 8$ measurements). Source data are provided as a Source Data file. In **b** and **c** are reported two typical spectra collected from AuNP in solutions at pH 6.0 and 7.4. Positions of the intensities related to bands A, B, and E are indicated by arrows and labels. The deconvoluted band C was used to normalize the A + B intensity. In acidic environments, the intensity of $COO^-$ symmetric stretching (A + B) decreases, reflecting the reduction of deprotonated carboxylic acid concentration. In alkaline solutions, the intensity of this vibration mode increases, especially sub-band B. Compared to band A + B, band E due to C = O stretching vibration in COOH is less intense and it has an opposite trend of intensity as a function of pH. Source dataSource data

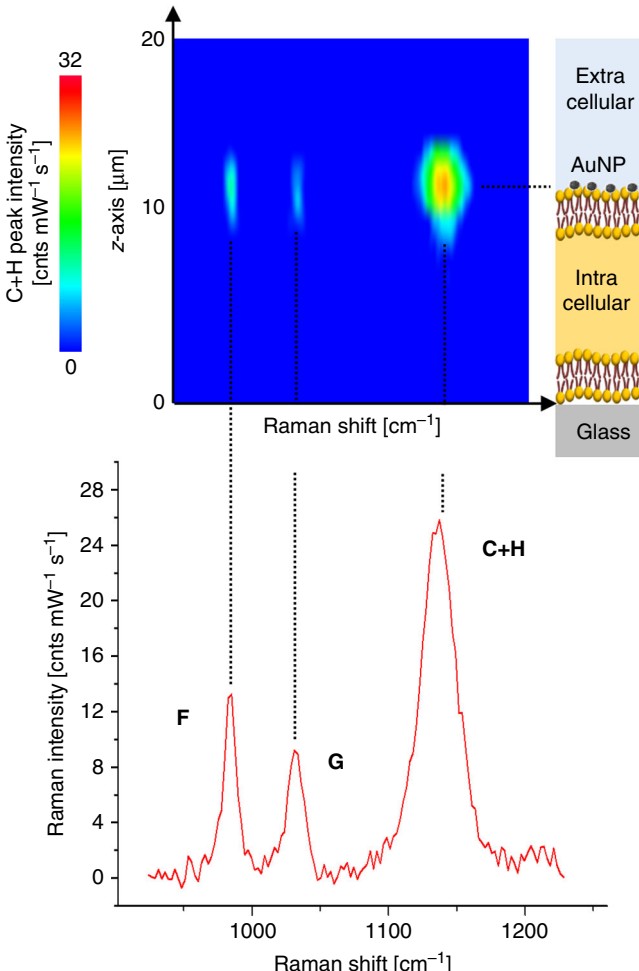

**Fig. 8** Visualization of AuNP anchored to the cell membrane by SERS. Typical results of SERS spectral intensity as collected from a z-axis line scan performed in the middle of a MKN28 cell after AuNP anchoring. The intensities in the range 925–1230 cm$^{-1}$ are shown as a function of the position along the z-axis. The z = 0 position corresponds to the bottom of the cell, which was identified by focusing the laser on the glass bottom dish. The symmetric shape of the SERS intensity distribution of each band along the z-axis confirms the location of the nanometer pH sensor in one single position, namely the point of maximum intensity at z = 12

function of time from EIPA addition showed the drop of the average intracellular pH from ~7.5 to ~7.3 (see Supplementary Fig. 6). Although these data were acquired using a conventional technique based on pH-sensitve fluorescent dye after cell internalization, they confirmed the limitation of proton extrusion induced by EIPA on MKN28 cells. Figure 9d reports an example of the results obtained from HepG2 after fixation, namely after cell death. The calculated pH was homogenously distributed on the surface and the average was equal to the pH of the buffer solution, with standard deviation of 0.1, which may be considered as an estimation of the standard error of this method. In Fig. 9f, we compared the mean values of pH calculated from three cells for each of the four investigated cases. The variations of pH in MKN28 cells after addition of EIPA and in HepG2 cells after fixation are both statistically meaningful (two-tailed unpaired $t$ test, $n = 3$, $p = 0.010$ and $0.001$, respectively). Such experimental evidence demonstrated that our pH nanosensor can visualize the dynamics of proton trafficking across the cell membrane with high spatial resolution. Supplementary Fig. 7 includes additional results of pH assessment performed on other cells. Figure 9g, h is example of spectra collected during the experiments on cells and representative of locations in which the pH was calculated as 6.9 and 5.4, respectively (see also Supplementary Fig. 8 for further examples).

## Discussion

The present investigation validated the strategy designed for high spatial resolution sensing of cell surface pH using SERS. Raman measurements enabled to visualize highly localized concentration of H$^+$ on the cell membrane, indicating the presence of steep pH gradients. The main peculiarity of this analytical method is the location of the nanosensor at nanometer distance from the membrane proteins, namely in proximity of ion channels and transporters. All the previously published results of pH measurement in cells using AuNP and SERS were obtained after endocytosis of the nanosensor, namely the calculated values were referred to pH in endosomes[22–26,50]. In some cases the lowest pH was reported as extremely acidic (i.e., as low as 3)[22], which may have been underestimated due to the bias introduced by considering the intensity of band H or I in the calculation of the band ratio. As stated in the introduction of this paper, the SERS-based method proposed by Sun et al.[18] enabled to visualize surface acidosis in HepG2, but the mean value of pH was about 7.3, namely much more alkaline than those reported here and in previous investigations on solid hepatocellular carcinomas[19–21].

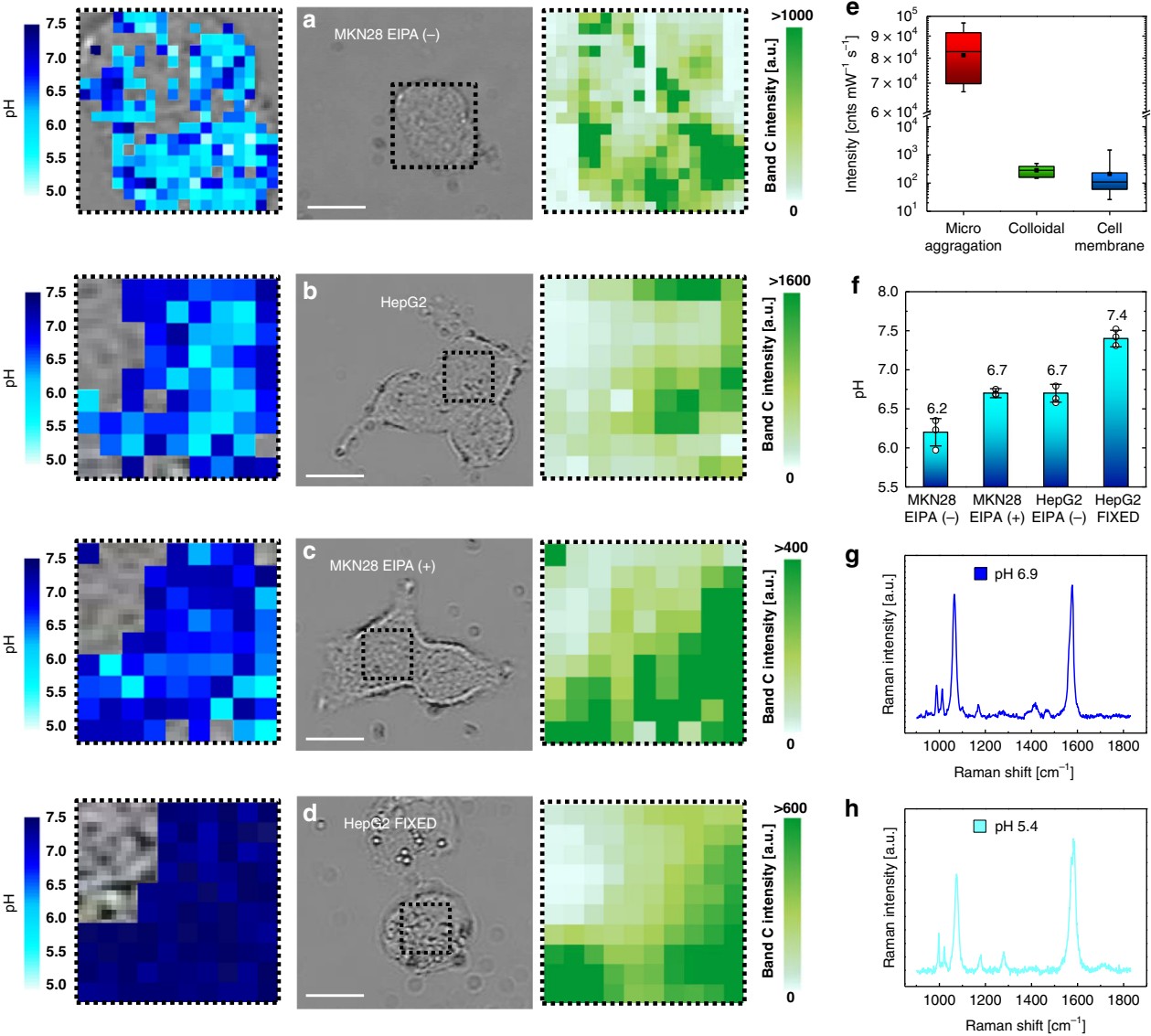

**Fig. 9** Results of cell surface pH analyses using SERS. **a–d** Examples of SERS hyperspectral pH mapping on MKN28, HepG2, MKN28 30 min after the application of EIPA and HepG2 after fixation, respectively. Each point of the maps corresponds effectively to a spot of 700 nm diameter. The relative images of Raman intensity of Band C at 1068 cm$^{-1}$ are also reported. Scale bars: 10 μm. **e** Box plot comparing the integrated intensities at 1068 cm$^{-1}$ collected from micro-aggregations ($n = 9$ measurements), AuNP colloidal solutions ($n = 9$ measurements) and AuNP on cell membrane surface ($n = 864$ measurements). The whiskers represent maximum and minimum; the square dot shows the mean and the segment in the rectangle indicates the median. **f** Mean values of surface pH in the four investigated cases (error bars show the standard deviation from the mean of $n = 3$ independent experiments). **g–h** Examples of SERS spectra collected from the outer surface of the cells and representative of pH 6.9 and 5.4, respectively. The band A + B was barely observable in the exemplifying spectrum of **h**. Source data are provided as a Source Data file. Source dataSource data

Although in the latter three studies hypoxia contributed to lower pH in the microscopic extracellular milieu of cancer tissues via the Pasteur effect, tumors acidosis occurs independently of hypoxia[19,51,52] and, accordingly, if the pH-sensitive probe on the substrate of ref. [18] had been indeed entirely attached to the cell surface, it should have detected higher proton concentration. HepG2 cells exhibit high concentration of filopodial protrusions formed on the membrane[53,54]. The TEM image in Fig. 3a clearly showed the presence of micrometric filopodia also on the basal surface of MKN-28 cells. From this morphological characterization it follows that in ref. [18] the cells adhered to the gold nanostructured substrate through filopodia, while an extended part of the 4-MBA SAM probed by the laser was presumably separated at micrometric distance from the membrane, where the effect of the buffer solution affected the pH results.

Conversely, the findings of our research are comparable to those obtained by means of low-pH insertion peptide functionalized with a pH-sensitive dye[12], since this technique has similar spatial resolution and location of the probe. According to that study, the average extracellular pH value near the plasma membranes of highly metastatic human HeLa was 6.4 ± 0.2, while in murine 4T1 mammary tumor cells was 6.1 ± 0.4. As in our experiments, both measurements were collected in the presence of glucose and they are consistent with the high level of acidification detected by our method in MKN28 and HepG2. Interestingly, in our analysis few locations of the membrane were characterized by pH 1.4 lower than the average. This remarkable difference may be explained by the highly localized anchoring of some nanometer SERS probes to points of proton extrusion, namely proteins of V-type H$^+$ ATPase and NHE.

In many spectra acquired on the cell surface we noticed the onset of the bands associated to the formation of decomposed aromatic species, TP. This phenomenon was also observed in a previous research based on SERS analysis after AuNP cellular uptake[25] and it may have been favored by enzymatic activity in physiological environment (e.g., ornithine decarboxylases secretion in case of MKN28[55]). Nonetheless, the reliability of the pH calculation proposed in this study is independent on the chemical–physical evolution of the 4-MBA SAM, since the fitting procedure enabled to discriminate the Raman intensity bands of pH-dependent moieties.

In conclusion, the proposed method of analysis can be used as a simple and viable tool to investigate and unfold the dynamics of proton exchange in cells, even after exposure to different pharmacological treatments or different physiological conditions. In addition, this protocol for unspecific labelling of outer cell membrane proteins can be adapted for the use of different types of gold nanosensor, which can vary in size, morphology and choice of SERS active compound. Recently, Carnevale et al.[56] developed an advanced method for intracellular pH measurement based on surface energy transfer using AuNP and fluorescence dyes, in which the rates of laser-induced photo-bleaching were reduced during multihour live-cell experiments. This method may certainly be extended to the measurement of cell surface pH by taking advantage of part of the knowledge introduced in our paper.

## Methods

**MKN28 and HepG2 cell cultures**. The moderately differentiated human gastric adenocarcinoma cell line, MKN28, was obtained from ATCC (Manassas, VA, USA). This cell line may be contaminated by MKN74 cells. Both MKN28 and MKN74 are gastric cancer cells with elevated glycolytic activity and expression of $Na^+/H^+$ exchanger, namely they are both suitable for testing the pH nanosensor presented in this study. We seeded the cells into 25 $cm^2$ flasks at a density of $2.5 \times 10^5$ cells/flask and incubated for 24 h (if not differently specified) in RPMI 1640 medium (Sigma-Aldrich, MO, USA) supplemented with 5% fetal bovine serum (FBS) in a humidified incubator at 37 °C with 5% $CO_2$ in air. The well-differentiated human hepatocellular carcinoma cell line, HepG2, was acquired from JCRB cell bank. We seeded the cells into 25 $cm^2$ flasks at a density of $2.5 \times 10^5$ cells/flask and incubated for 24 or 72 h in DMEM medium (Sigma-Aldrich, MO, USA) supplemented with 10% FBS in a humidified incubator at 37 °C with 5% $CO_2$ in air.

**Conjugation of AuNP for SERS and CFM analyses**. As gold nanosubstrates, we selected the commercially available 90 nm gold nanourchins in 0.1 mM PBS (Sigma Aldrich, St. Louis, Missouri, USA). EZ-link-HPDP-Biotin (Thermo Scientific, MA, USA) is a N-(6-(Biotinamido)hexyl)-3′-(2′-pyridyldithio)-propionamide compound (HPDP-B), which was dissolved in DMSO to prepare 10 μM stock solution. The pH-sensitive 4-MBA compound is the 4-mercaptobenzoic acid (Sigma Aldrich, MO, USA), which was dissolved in EtOH (1 mM stock solution). The nanosensor for pH assessment by SERS was prepared by adding 50 μL 4-MBA + 5 μL HPDP-B (concentration ratio 1000:1) to 1 mL of AuNP colloidal solution, which was incubated for 30 min at 21 °C using a rotating mixer. Separation of AuNP was obtained by 30 min centrifugation at 250g, which was repeated on the supernatant for 30 min at 1000g. During the spectroscopic analyses of cells we dispersed the AuNP in isotonic buffer solution containing 115 mM NaCl, 15 mM NaNO$_3$, 5 mM KCl, 1 mM Ca(NO$_3$)$_2$, 1 mM Mg(NO$_3$)$_2$, 10 mM HEPES, 10 mM glucose, 10 mM sucrose, and adjusted to pH 7.4 by adding CsOH. For the calibration of the nanosensor, the AuNP were resuspended in isotonic buffer solutions at different pH (i.e., in the pH range 4–11). In the case of CFM experiments, the fluorescent dye for AuNP conjugation is the AlexaFluor®488-streptavidin (Thermo Scientific, MA, USA), which was dissolved in 0.01 M PBS to prepare 10 μM stock solution (referred as Alexa-SA). For fluorescence microscopy, 1 mL AuNP colloidal solution was conjugated with 15 μL HPDP-B and mixed for 30 min at RT. After separation by double centrifugation, we resuspended the AuNP in 0.1 mM PBS and we added 10 μL Alexa-SA. The final steps were incubation for 30 min at 21 °C, separation by double centrifugation and resuspension of the conjugated AuNP in 1 mL buffer solution at pH 7.4. The HPDP-B:AlexaSA ratio was about 3:2 in order to keep unreacted biotin on the surface of AuNP.

**Protocol for cell surface labelling with conjugated AuNP**. Cells seeded on glass bottom dishes were placed at 4 °C for 10 min and rinsed twice with ice-cold 0.01 M PBS + 2 mM CaCl$_2$/1 mM MgCl$_2$ at pH 7.4 (hereafter referred as PBS-2Ca). EZ-link Sulfo-NHS-SS-Biotin (100 mg powder, Thermo Scientific, MA, USA) is the

sulfonated N-hydroxysulfosuccinimide-esters of biotin (NHS-B) used as surface protein biotinylation reagent, since it is not membrane permeable due to the presence of the charged sulfo group. We treated each dish with 1 mg NHS-B in 1 ml PBS-2Ca at pH 8.0 for 30 min at 4 °C. After rinsing with ice-cold PBS-2Ca (3 times per 3 min), we treated the cells with 200 μl of 10 μM streptavidin (Sigma Aldrich, MO, USA) in 1 ml PBS-2Ca at pH 7.4 for 30 min at 4 °C. Following rinsing in PBS-2Ca, the cells were incubated with 1 ml of conjugated AuNP solution for 15 min at 4 °C followed by 15 min at 21 °C. Robust washing with 0.01 M PBS for five times was the final step before SERS or CFM analysis. During the spectroscopic analyses we added the isotonic buffer solution described in *Conjugation of AuNP for SERS and CFM analyses*. Isotonic buffer solution containing 100 μM ethylisopropyl amiloride (EIPA, Sigma Aldrich, MO, USA) was used to study the effect of the NHE inhibitor 30 min after addition. In case of CFM analysis the cells were also stained with nuclear dye Hoechst 33342 (Thermo Scientific, MA, USA).

**Protocol for cell surface labelling with NHS-B and Alexa-SA**. For confirmation and optimization of surface biotinylation by NHS-B, MKN28 cells were rinsed twice with ice-cold PBS-2Ca at pH 7.4 and treated with 1 mg NHS-B in 1 mL PBS-2Ca for 30 min at 4 °C. We prepared 4 series of 3 samples using PBS-2Ca at pH 7.4, 7.6, 7.8, and 8.0. After rinsing in ice-cold PBS-2Ca 3 times for 3 min, we fixed the cells with 2.5% glutaraldehyde + 2% paraformaldehyde (PFA) in PBS (1 ml/well). After 30 min at 21 °C, we rinsed the samples in PBS 3 times for 3 min and we treated the samples with 20 μL Alexa-SA in 0.5 mL 0.01 M PBS. Finally, after 20 min the samples were rinsed 3 times for 5 min in PBS, stained with nuclei dye Hoechst 33342 (Thermo Scientific, MA, USA) and analyzed by CFM.

**Confocal fluorescence microscopy**. We collected images of fluorescence intensity using the LSM 510 META confocal laser scanning microscope equipped with a C-Apochromatic 40×/1.2 W objective lens (Carl Zeiss, Jena, Germany). The fluorescence dyes were excited either at $\lambda = 780$ nm using a Ti:Sapphire laser (MaiTai®, Spectra-Physics, CA, USA) via the two photon absorption process or at $\lambda = 488$ nm using a Ar$^+$ laser (Ar-ion, LASOS Inc., Germany) via the single photon absorption process. Using band-pass filters, the emitted fluorescence was detected in the range 390–465 nm for Hoechst 33342 and in the range 500–530 nm for Alexa488. The z-stack imaging enabled to reconstruct 3D plots of fluorescence intensity distribution: xy images were collected with the confocal pinhole of 70 μm (1 μm depth of focus) at different positions of the focal plane along the z-axis, from the glass bottom up to above the cells with 1 μm step. Data processing was performed using the software ZEN 2012 (Zeiss, Jena, Germany) and ImageJ (U.S. National Institutes of Health, Bethesda, Maryland, USA).

In addition, we monitored the variation of cytosolic pH in MKN28 cells treated with EIPA using carboxy-seminaphthorhodafluor-1 (carboxy-SNARF-1) (Molecular Probes, Eugene, OR, USA), a pH-sensitive fluorescent dye. The fluorescence dyes were excited at $\lambda = 514$ nm and we collected fluorescence light centered at 645 and 592 nm. The emission ratio (645/592 nm) was calibrated using calibration buffer solutions at different pH. Several regions of interest with a diameter of 1 μm were then randomly selected on MKN28 cells, excluding nuclear regions. The fluorescence emission ratio was calculated and used to estimate pH from the calibration curve.

**TEM imaging**. TEM images were collected from 90 nm slices of cells embedded into epoxy resin after AuNP attachment, fixation and dehydration. The MKN28 cells were seeded onto plastic coverslips (Nunc™ 174950, Thermanox™ Cell Culture Cover Slips), which were used in 24-well plates for 72 h incubation in culture medium. Following the completion of the protocol for membrane surface labelling with pH-sensitive AuNP, we waited 1 h before fixing the cells with 2.5% glutaraldehyde/2% paraformaldehyde (PFA) in 0.1 M PBS for 30 min at 21 °C (0.25 ml/well), followed by postfixation with 1% osmium tetroxide containing 1% potassium ferrocyanide in 0.1 M PBS at 21 °C for 1 h. Both fixation and postfixation steps included rinsing in ultra-pure water. Dehydration was performed by dipping the sample into EtOH solutions at different concentrations (i.e., 30, 50, 70, 90, 100, and 100%) for 10 min each. Dehydrated samples were immersed into half epon (50: 50 resin: EtOH mixture) for 30 min followed by immersion into full epon for 1 h twice. Finally, the solid pellet was obtained by embedding the cells into epoxy resin (Poly/Bed 812, Polysciences, Inc., Warrington, PA, USA) at 60 °C overnight. Ultrathin sections of the pellet containing cells were trimmed using an ultra-microtome (Reichert Ultracut S, Leica, Wetzlar, German) equipped with a diamond knife. The ultrathin sections were subsequently collected onto Formvar-coated grids and stained with EM stainer (Nisshin EM, Tokyo, Japan) and Reynold's lead citrate. We examined the cross-sections by using an H-7100 transmission electron microscope (Hitachi, Tokyo, Japan) operating at 75 kV. The images were taken at different magnifications (7, 17, and 50 k) and recorded with a CCD camera (C4741-95; Hamamatsu Photonics) at a pixel size of 9.43 nm.

**SERS analysis**. Spectra were collected using a Raman microscope (Raman-11, Nanophoton, Osaka, Japan), equipped with a 671 nm laser excitation source (Ignis, Laser Quantum, Manchester, UK) and a 60× water immersion objective lens of NA = 1.1 (Olympus, Tokyo, Japan). Before reaching the CCD, the scattered light was diffracted utilizing a 600 grooves/mm grating. The in-plane spot size and the axial

resolution of the laser probe using a cross-lit aperture of 70 μm were estimated performing experiments with fluorescent beads of 50 nm diameter on a silica glass immersed in deionized water. We collected the light emitted during $x$–$y$ plane and $y$–$z$ plane raster scans with a step of 100 nm. The plots of the intensity at $\lambda_e = 690$ nm (i.e., the emission wavelength of the fluorescent bead) enabled us to visualize their positions in the $x$–$y$ and $y$–$z$ planes. The intensity profile given by a line of points passing through one bead along the $x$-axis represents the Gaussian shaped laser probe response in the focal plane. The width of the Gaussian function at $1/e^2$ of its peak can be considered as an estimation of the spot size, which was about 700 nm. The intensity profile obtained from a line of points passing through one bead along the $z$-axis represents the Lorentzian shape axial probe response. The FWHM was estimated equal to 3.1 μm. During calibration experiments of 4-MBA/HPDP-B conjugated AuNP at different pH, the laser power was set at 20 mW with acquisition time of 20 s for each spectrum. Some spectra from colloidal solution were also acquired at 1 and 72 mW to study the influence of the laser power. During the analysis of cells, we reduced the laser power to 10 mW for 5 s. Spectral Raman lines were analyzed and deconvoluted using a commercially available software package (Origin 9.1, OriginLab Co., MA, USA). Fitting of intensity bands was performed using Gaussian-Lorentzian (i.e., Voigtian) functions after baseline subtraction.

**EF calculation**. We estimated the average EF of the conjugated AuNP by collecting normal Raman scattering (NRS) from a 200 mM 4-MBA ethanol solution and SERS intensities from 1 μM 4-MBA colloidal solution of AuNP. Supplementary Fig. 9a shows examples of NRS and SERS spectra collected from the two solutions. The NRS spectra were obtained after subtraction of the ethanol bands collected from pure solvent under same experimental conditions, as shown in Supplementary Fig. 9b. The following formula was used to calculate EF:

$$EF = \frac{I_{SERS} \times N_{NRS}}{I_{NRS} \times N_{SERS}}, \tag{2}$$

where $I_{SERS}$ and $I_{NRS}$ are the average intensities of the $\nu_{12}$ aromatic ring vibration from SERS and NRS spectra at 1068 and 1094 cm$^{-1}$, respectively; $N_{SERS}$ and $N_{NRS}$ are the number of molecules in the laser probe volume during SERS and NRS measurements. Since the experiments were carried out using the same objective lens and the same confocal slit aperture (i.e., equal laser probe volume), $N_{SERS}$ and $N_{NRS}$ can be substituted by the 4-MBA concentration in the solutions for SERS and NRS experiments, respectively. From $n = 10$ measurements for each solution, $I_{SERS}$ and $I_{NRS}$ were determined to be 284 and 34 cnts mW$^{-1}$ s$^{-1}$, leading to an average EF equal to $1.67 \times 10^6$. This value is lower-bound estimate, since the SERS signal is predominately associated to the 4-MBA molecules in the hot spots localized on the AuNP surface.

**Cytotoxicity assay**. MKN28 and HepG2 cells were seeded onto a 96-well plate and incubated for 48 h. For each cell type, eight wells were treated for AuNP anchoring to the outer membrane surface and eight wells were used as control. Since the protocol presented in this study includes several rinses of the cells, we performed the same number of rinses also in the control wells. Cell viability was measured by MTT assay. Briefly, 5 mg of MTT were dissolved in 1 mL isotonic buffer solution and in each well 200 μL were added for 3 h. Then buffer solution was replaced with DMSO to dissolve blue formazan crystals, and plate was shaken for 15 min in the dark. Results were obtained by measuring the absorbance at 570 nm using an automated micro plate reader (SpectraMax M2e, Molecular Devices, San Jose, CA, USA). We performed three independent experiments. For each cell type, we calculated the average percentage of cell viability, assuming the average viability of control cells as 100%. Further analysis of cytotoxicity was performed by red-fluorescent PI and blue-fluorescent Hoechst 33342 staining on cell seeded on 35 mm glass bottom dishes with and without AuNP. The fluorescence microscopy images were collected at least after 1 h from the completion of the AuNP attachment protocol.

**Statistical analysis**. In the case of pH assessment on the outer cell membrane surface, the data compared for each group (i.e., MKN28 control, EIPA-treated MKN28, HepG2 and HepG2 after fixation) represent the mean of $n = 3$ independent experiments ± SD. For each cell type, differences between two groups were statistically validated using two-tailed unpaired $t$ test ($p < 0.05$). The same statistical test was also used to verify the difference in cell viability between control and AuNP treated samples, for each cell type.

## Data availability
The data that support the findings of this study are available from the corresponding authors upon reasonable request.

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

## Acknowledgments

We thank Emeritus professor Yasunobu Okada for the comments and the enlightening discussions. This work was supported by a Grant-in-Aid for Young Scientists (B) of The Japanese Society for the Promotion of Science (JSPS KAKENHI Grant Number 16K18990 (L.P.)).

## Author contributions

L.P., Y.M., and S.H. conceived and designed this study. L.P., S.H., H.S. prepared the samples for Raman and CFM experiments. L.P., S.H., and Y.K. performed Raman and CFM experiments. L.P., S.H., H.S., K.M., and T.S. prepared the samples for and performed TEM analysis. Y.M., T.I., and H.T. contributed to the experiments and the development of the experimental protocol. L.P. wrote the manuscript text. L.P., S.H., and Y.K. prepared figures. All authors reviewed the manuscript.

## Additional information

**Competing interests:** The authors declare no competing interests.

