## [Peer Review File · Nature Communications]

Reviewers' comments:

Reviewer #1 (Remarks to the Author):

In this manuscript, the authors described a surface enhanced Raman spectroscopy (SERS) approach to monitor pH on cell surfaces with a relatively high spatial resolution. AuNPs modified with 4-mercaptobenzoic acid (4-MBA) were conjugated on cell surfaces using the biotin-streptavidin interaction. The monolayer of 4-MBA on AuNPs has strong surface plasmon enhanced Raman scattering, and the scattering depends on the pH of the surrounding nanoenvironment, enabling the sensing of local pH on cell surfaces. This study represents a new application of cell membrane modification and is of interest to the field. The developed method may be useful to reveal some fundamental biological questions. However, there are several major issues need to be addressed.

1. Figure 2c-d showed nice ring like structures, indicating Alexa-streptavidin was on the cell membranes. However, the fragmentation of nuclei may be an indicator of cell apoptosis. The data raise a concern on the viability of cells after the treatment. The authors claimed that the modification did not induce cell death, but there is no any cell viability data in the main text or supporting information.
2. In this study, cell membranes were first biotinylated and conjugated with biotinylated AuNPs using streptavidin as the bridge. The same method has been previously developed for anchoring NPs on cell membranes (Cheng et al. *ACS Nano*, 2010, 4, 625), in which some data may be used as references for works presented in this manuscript. A review paper (Stephan et al. *Nano Today*, 2011, 6, 309.) may be helpful for the authors to learn more about engineering cell surfaces with NPs. The authors have claimed that their AuNPs stayed on cell surfaces without detectable internalization by MKN28 cells. It is important to describe clearly the time frame of the study because NPs cannot stay on cell surfaces forever. To explain the low NP uptake, the authors mentioned that "the sulfo group in the molecular label prevented endocytosis" on page 5 without citing any references. Even if the free linker molecule is cell membrane-impermeable, it cannot explain the low endocytosis of AuNPs at here because the endocytosis of AuNPs likely depends on the cell membrane proteins that AuNPs linked to. In the previous paper of Cheng et al, it was explained that the multiple binding of NPs with cell membranes interfered with cell endocytosis and caused NP aggregation on cell membranes.
3. To conjugate AuNPs on cell membranes, streptavidin was attached on bionylated AuNPs, but leaving some biotin sites available. The authors then biotinylated cell membranes and coated another layer of streptavidin on cells before conjugating AuNPs on cell surfaces. The rationale of this complicated process is unclear. Why don't directly conjugate streptavidin-coated AuNPs on biotinylated cell membranes or directly conjugate biotinylated AuNPs on streptavidin-coated cell surfaces?
4. The authors have used the term pH nanosensor. However, the resolution of SERS signal was about 1 μm as shown in Figure 7. The 2D SERS signal itself cannot directly distinguish protons in cytosol of cells and on cell membranes. It relies on the navigation of AuNPs via other methods. Compared to a recent publication utilizing surface energy transfer to AuNPs for local intracellular pH detection (ACS Nano DOI: 10.1021/acsnano.8b02200), the approach described in this manuscript does not seem have advantages in spatial resolution or convenience for broad applications. Note the method in this very recent ACS Nano paper can be extended to membrane-conjugated AuNPs.
5. Figure 7c studied the change of average pH value on cell surfaces with and without the presence of EIPA. To further validate the strategy, an established method detecting the pH variation caused by EIPA should be used as a control.
6. The TEM images in Figure 4 clearly showed AuNP aggregation and a heterogeneous distribution on cell surfaces. It is unclear if the local SERS signal difference is due to pH variation on cell surfaces or the difference of local AuNP density. An experiment needs to be designed to address this question. For example, fixing the cells after AuNP conjugation on cell surfaces may eliminate local pH difference. Can the SERS signal become homogeneous?

Reviewer #2 (Remarks to the Author):

In this work, authors reported a method for interstitial pH sensing with high resolution based on SERS. Specifically, they immobilized the gold nanoparticles (AuNPs), which are functionalized with 4-mercaptobenzoic acid (4-MBA), to the cell membrane via a streptavidin bridge using two molecular compounds containing biotin: a pyridyldithiol-biotin compound (HPDP-B), which conjugates to AuNP via thiol-gold interaction, and a sulfo-NHS-ester-biotin compound (NHS-B) that reacts with the primary amines of lysine and the amino-termini of polypeptides on the outer cell membrane surface. Since the SERS signal of the COO⁻ band varies with pH values due to protonation/de-protonation, the relative peak intensity change is used to indicate the pH values after obtaining the calibration curve. Using 4-MBA functionalized SERS-active substrates (either AuNPs (refs 19-22) or Au nanostructures (e.g., *Biosensors and Bioelectronics* 73 (2015) 202–207) for cell pH sensing have been reported. From this point of view, this method is not new. The unique of this work lies on the using of NHS-B to un-specifically link to membrane proteins and then using streptavidin to further link HPDP-B to 4-MBA functionalized AuNPs. In this way, AuNPs are located at the outside of cell membrane close to ion channels and transporters, enabling pH sensing locally and even after treated with NHE inhibitor, while AuNPs are located in the intracellular in the previous reported works (refs. 19-22). However, the work (*Biosensors and Bioelectronics* 73 (2015) 202–207) reported the direct and accurate mapping extracellular pH of living cells in complex media using SERS based on 4-MBA functionalized Au nanostructure arrays. Therefore, in general, the work is an increment, lacks of novelty. In addition, more comments are provided below.

1. The assignment of 1068 and 1588 cm⁻¹ peaks in the sentence on P.6 line 207 is wrong, i.e., "...bands located at 1068 and 1588 cm⁻¹ are due to v_{8a} and v₁₂ aromatic ring vibrations, respectively,..." The bands located at 1068 and 1588 cm⁻¹ should be due to v₁₂ and v_{8a} aromatic ring vibrations, respectively. Therefore, the sentence on p. 7 line 229 "...v_{8a} or v₁₂ aromatic ring vibrations, namely bands C+H or D+I in Fig. 5 (a),..." is also wrong.
2. The assignment and explanation about 1390 and 1415 cm⁻¹ bands could also be questionable. In the work reported in *Biosensors and Bioelectronics* 73 (2015) 202–207, it showed the peak intensity change of two peaks versus the pH values from 4.05 to 8.99. In that work, Au nanostructure instead of AuNPs was used as the SERS-active substrates, so there is less or no chance for 4-MBA molecules that have form SAM to undergo a dramatic orientation change.
3. Caption of Figure 5a should provide the ratio of 4-MBA/HPDP-B. The 2-pyridine shown in Figs. 5a and 5b could cause a confusion as there is no 2-pyridine was immobilized on AuNPs. Why are the four strong peaks shown in Fig. 5b due to HPDP-B related to 2-pyridine? Due to the smaller Raman scattering cross-section of HPDP-B compared to 4-MBA, it is reasonable that the intensity is about one order of magnitude lower than 4-MBA in Fig. 5a. The four strong peaks in Fig. 5b should be more related to O=C-NH as they are more close to sensing surface.
4. As reported in *Biosensors and Bioelectronics* 73 (2015) 202–207, other cations, especially divalent cations (Ca²⁺, Mg²⁺) can cause the intensity change and the peak shift of ~1420 cm⁻¹ peak. In this study, as stated on p.11 lines 358-361, "During the spectroscopic analyses we used an isotonic buffer solution containing 115 mM NaCl, 15 mM NaNO₃, 5 mM KCl, 1 mM Ca(NO₃)₂, 1 mM Mg(NO₃)₂, 10 mM HEPES, 10 mM glucose, 10 mM sucrose and adjusted to pH 7.4 by adding CsOH." Therefore, the calibration curve (Fig. 6a) should also be done in such solution. While this won't affect the trend by adding EIPA, the real pH values could be different.
5. The 2D pH mapping via the xy raster scan shown in Figure 7 should provide the z position. As

shown in Figure 2 of Supporting Information, SERS spectra signal only appeared locally where AuNP aggregations are. Also, for Figure 2 of Supporting Information, it also needs to indicate where this z-scan was done. My guess is that this should not be in the middle of the cell as the strong SERS signal from the middle of the z-scan range. This z-scan might be taken close to the edge of the cell.

6. The claim of two peaks at 999 and 1021 cm^{-1} , which are associated with the decomposed aromatic species (i.e., TP), needs to be careful. Why there are no obvious such two peaks in the pH calibration experiments, i.e., Figure 6, for both acidic (pH = 6.0) and basic (pH = 8.0), compared to the detection of cells (Fig. 7)? The increase of these two peaks could be due to the orientation change of 4-MBA. The increase of these two peaks upon the binding of fructose on 4-mercaptoproic acid SAM was reported in ACS Nano 9, 2668 (2015).

7. In the Figure 3 of Supporting Information, the optical images show clearly that there are many AuNPs at the edges of cells. Did authors collect the SERS data from those areas? How about the pH values of those areas?

8. CFM was applied to confirm the label was only at the membrane of cells. However, as shown in TEM and even optical images, the AuNPs can be clearly seen. Therefore, CFM method may not be needed.

9. Figure 1 of Supporting Information needs to provide the z values for each line scan.

10. The wavenumber of peak around 1600-1670 cm^{-1} was labeled differently in the Figure 7 caption (1063 cm^{-1}) but 1068 cm^{-1} in the legend.

Reviewer #3 (Remarks to the Author):

In the present article a novel method (or approach) that aims to measure the extremely sharp pH gradients in extracellular environments, close to cell membrane proteins, is developed. This is a a topic of great relevance in Cell Biology due to its critical role in controlling many cellular functions, including cell proliferation, metastasis, drug resistance and apoptosis.

The method consist in anchoring Au nanoparticles (NPs) functionalized with a mercaptobenzoic acid (MBA) to the outer part of the cell membrane. As the MBA molecule is known to be a molecule whose Raman response is dependent on the external pH, a calibration curve that relates the ratio between typical Raman bands of this molecule and the pH could be constructed. The presence of Au NPs provides an enhancement of the Raman signals of MBA (SERS) as well as a method to localize them only on the extracellular membrane region.

This attachment is achieved by treating the membrane with a sulfo-NHS-ester-biotin compound which reacts specifically with the polypeptides present in the membrane and by functionalizing the Au NP also with a pyridyldithiol-biotin compound (HPDP-B), and then adding Streptavidin (STV) that acts as a bridge between the biotinylated Au NP and the biotinylated cell membrane.

At the same time control confocal fluorescence microscopy (using STV with a fluorescence probe molecule) experiments, as well as TEM images, demonstrate the successful functionalization of the external part of the cell membrane.

All the experiments were performed with human gastric adenocarcinoma cell line, MKN28, and the shape of Au NPs (nano-urchins) was chosen to maximize the SERS enhancement of the MBA molecule.

The nanometer size of the AuNP sensor attached to the proteins and the use of SERS allows to visualize the highly localized variation of pH induced by proton extrusion, which is upregulated in cancer cells. The measurements were performed before and after addition of ethyl-isopropyl amiloride (EIPA), an inhibitor of Na⁺/H⁺ exchanger (NHE). In addition Raman image was also performed for scanning (using the MBA molecule as reporter) the pH variation along the z direction.

The article is well written and of great interest for a wide audience , however it seems to me that the following issues should be addressed, before the manuscript should be suitable to be published in Nature communications:

a) For the calibration curve, the SERS active modes coming from HPDP B and TP molecules seems to be taken into account. However in the real samples also the Raman bands of Streptavidine should experience significant SERS enhancement.

Have the authors considered this issue?

b) The resolution of the average pH measurements are diffraction limited to the size of the laser spot, therefore the SERS signals are the average within this area and also the pH values. It is not clear why the axial resolution is diffraction limited while the resolution along the other directions is dictated by the size of the Au NP degree of agglomeration. This point should be explained more clearly.

c) Can the author estimate the SERS enhancement factor of the relevant molecular probe involved (MBA) and its dependence on the degree of Au NP agglomeration. This issue is very important since the value of pH depends on the SERS intensity of the relevant bands and their intensities should also depend on the degree of agglomeration and the average distance between Au NPs.

**Response to the Reviewers' comments and remarks:**

We sincerely thank the Reviewers for their time and effort in reviewing our paper. Independently on the final
result of this submission, we would like to express our appreciation and gratitude for the extremely valuable
and pertinent comments to our research. We strongly believe that the Reviewers' remarks and opinions
highly contributed to improve the quality of our work after thoughtful and careful revision. In the revised
version of the manuscript, we addressed all the queries raised by the Reviewers, point by point. In addition,
new data are also presented to elevate the quality of our study, which is now much more detailed and
comprehensive. We are confident that the new compelling experimental evidence and discussion will remove
any doubt over our statement of unprecedented method for cell surface pH measurement based on surface
enhanced Raman spectroscopy. The revised parts in the new manuscript and the answers to each comment in
this document are written in red:

Reviewer #1 (Remarks to the Author):

1. Figure 2c-d showed nice ring like structures, indicating Alexa-streptavidin was on the cell membranes.
However, the fragmentation of nuclei may be an indicator of cell apoptosis. The data raise a concern on the
viability of cells after the treatment. The authors claimed that the modification did not induce cell death, but
there is no any cell viability data in the main text or supporting information.

*In the revised manuscript we provide cell viability data (MTT assay and propidium iodide (PI) fluorescent*
*staining) as requested by the Reviewer. Please refer to the new text added at the end of the section* *Outer cell*
*membrane surface labelling* (pg. 6, starting from "As a final validation step of the protocol,..."); the new Fig.
4 and the details of the protocol at pg. 18, in the paragraph *Cytotoxicity assay* of the section *Methods*. We
acknowledge the importance of this information that was missing. Please note that Fig. 2 of the original
manuscript is Fig. S1 in the revised manuscript (it was moved to *Supporting information*). We would like to
remark the fact that in Figs. S1 (e)-(g) nuclear fragmentation can be observed in two cells that are in the final
stage of mitosis, during which nuclear fragmentation is a physiological feature. In order to substantiate this
point, we also provide new fluorescence images obtained from MKN-28 cells stained with PI and Hoechst
dyes, in which there are clear examples of mitosis with nuclei fragmentation and without PI red fluorescence.
Please refer to Figs. (f)-(i). We believe that the presented new images represent a further validation of the
cell viability test results, namely the lack of cell apoptosis induced by our treatment.

2. In this study, cell membranes were first biotinylated and conjugated with biotinylated AuNPs using
streptavidin as the bridge. The same method has been previously developed for anchoring NPs on cell
membranes (Cheng et al. ACS Nano, 2010, 4, 625), in which some data may be used as references for works
presented in this manuscript. A review paper (Stephan et al. Nano Today, 2011, 6, 309.) may be helpful for
the authors to learn more about engineering cell surfaces with NPs. The authors have claimed that their
AuNPs stayed on cell surfaces without detectable internalization by MKN28 cells. It is important to describe
clearly the time frame of the study because NPs cannot stay on cell surfaces forever. To explain the low NP
uptake, the authors mentioned that "the sulfo group in the molecular label prevented endocytosis" on page 5
without citing any references. Even if the free linker molecule is cell membrane-impermeable, it cannot
explain the low endocytosis of AuNPs at here because the endocytosis of AuNPs likely depends on the cell
membrane proteins that AuNPs linked to. In the previous paper of Cheng et al, it was explained that the
multiple binding of NPs with cell membranes interfered with cell endocytosis and caused NP aggregation on
cell membranes.

*We thank Reviewer #1 for the interesting comment and suggestions. We referred the work by Chen et. al. at*
*pg. 4, Line 6 from the bottom (Ref. [28]). We humbly admit that we were not aware of that work and we*
*believe that the previous use of NHS-biotin for unspecific labeling of surface proteins with Neutravidin*
*conjugated nano-fluorospheres should be mentioned, but it does not diminish the novelty of our pH sensor.*
*In fact, the basic idea of our method has a different purpose and, consequently, the strategy for nanoparticles*
*conjugation and attachment is more elaborated. The use of NHS-biotin and streptavidin (SA) is only one*
*piece of a more complex puzzle that had to be carefully assembled to achieve the final goal of attachment*

AND pH measurement. The AuNP surface cannot be entirely conjugated with SA as reported by Chen et. al.
because the SERS intensity from SA should not interfere with the 4-MBA signal. For this reason, we firstly
conjugated SA to the NHS-biotin attached to the outer membrane proteins and then with added the AuNPs
functionalized with 4-MBA and HPDP-biotin. The result is approximately 1 SA for 1 AuNP and,
consequently, the effect of this protein to the collected SERS signal is negligible. For the same reason, also
the concentration of HPDP-biotin bound to AuNP had to be adjusted to avoid contribution to the collected
SERS intensity. The result of this calibrated procedure is the homogeneous distribution of AuNP showed in
our paper and the formation of nanoclusters. We believe that these arguments require to be clearly elucidated
in the revised manuscript and we took advantage of the Reviewer's comment to implement the text at the
beginning of paragraph *Outer cell membrane surface labelling* at pg. 4. As far as the AuNP internalization
issue is concerned, when we mention that "the sulfo group in the molecular label prevented endocytosis" on
page 5, we are referring to the previous Fig. 2 (now S1 of *Supporting information*), which reports the results
of cell surface biotinylation without AuNP. In fact, the experiments with sulfo-NHS-biotin conjugated with
fluorescent dye (i.e., without AuNPs) were designed to study the reaction of NHS ester group to the cell
membrane proteins. In cancer cells we found that higher pH of the buffer solution during 30 minutes of
biotinylation increased the yield of reaction. These experiments also confirmed the homogeneous distribution
of sulfo-NHS-biotin molecules attached to the cell surface. The presence of the sulfo-group prevents
internalization of NHS-biotin and enables only the selective targeting of cell outer membrane proteins. Once
the AuNP are attached to NHS-biotin after the following step of the protocol, we are completely aware that
the endocytosis of AuNP is not prevented by the sulfo-groups. We apologize for the improper use of
"endocytosis" in the phrase mentioned by the Reviewer, which may have been misleading. In the revised
manuscript we changed the word "endocytosis" to "internalization", which is referred only to NHS-biotin
during the biotinylation step (see pg. 5, 7th line from the bottom of the page). The lack of AuNP endocytosis
during our Raman experiments was proved by the results showed in Fig. 3 (i.e., Fig. 4 of the previous
version): the TEM images were purposely taken from cells that underwent to fixation 1 hour after the end of
AuNP attachment, which corresponds to the average time spent for the Raman experiments shown in this
paper. In the original manuscript, at Line 19 of pg. 5, we stated that we waited 1 hr before fixation and TEM
imaging. Cheng et al. showed lack of endocytosis for up to 2 days, which is a remarkable result that was
explained by nanoparticles binding with multiple surface proteins. In the revised version we added this
explanation in the new text from Line 7 of pg. 6. In addition, Reference [30] (Stephan et al. Nano Today,
2011, 6, 309) was added at pg. 6, Line 13.

3. To conjugate AuNPs on cell membranes, streptavidin was attached on bionylated AuNPs, but leaving
some biotin sites available. The authors then biotinylated cell membranes and coated another layer of
streptavidin on cells before conjugating AuNPs on cell surfaces. The rationale of this complicated process is
unclear. Why don't directly conjugate streptavidin-coated AuNPs on biotinylated cell membranes or directly
conjugate biotinylated AuNPs on streptavidin-coated cell surfaces?

Actually, we directly conjugated biotinylated AuNPs to streptavidin-coated cell surfaces. In the paragraph
*Conjugation of AuNP for SERS and CFM analyses* of the section *Method*, we did not mention any incubation
of AuNP with SA. The nanoparticles are conjugated with 4-MBA and HPDP-biotin. In the paragraph
*Protocol for cell membrane surface labelling with conjugated AuNP* we explained 3 separated steps: 1)
NHS-Biotin reacting with cell surface proteins; 2) washing and addition of SA that reacts with NHS-biotin;
3) washing and incubation with 4-MBA/HPDP-biotin conjugated AuNP. Please see the new text at the
beginning of paragraph *Outer cell membrane surface labelling* (pg. 4), as also already discussed in the
answer to Comment 2.

In the case of CFM experiments, we conjugated AuNP with streptavidin-Alexa488 in order to attach the
fluorescent dye to some of the HPDP-biotin previously attached to the AuNP. The HPDP-biotin:streptavidin-
Alexa488 ratio was 3:2, in order to leave unreacted biotins on the surface of AuNPs for the attachment to
surface membrane proteins coated with streptavidin.

4. The authors have used the term pH nanosensor. However, the resolution of SERS signal was about 1 μm
as shown in Figure 7. The 2D SERS signal itself cannot directly distinguish protons in cytosol of cells and on
cell membranes. It relies on the navigation of AuNPs via other methods. Compared to a recent publication
utilizing surface energy transfer to AuNPs for local intracellular pH detection (ACS Nano DOI:
10.1021/acsnano.8b02200), the approach described in this manuscript does not seem have advantages in

spatial resolution or convenience for broad applications. Note the method in this very recent ACS Nano
paper can be extended to membrane-conjugated AuNPs.

We believe that the Reviewer is referring to 1 μm as the in-plane resolution. The in-plane resolution is given
by the size of the laser spot, which is diffraction limited and it was estimated as 700 nm (See the new Fig. 3
(b) in the revised manuscript). Note that in the original version we erroneously reported the FWHM of the
Gaussian beam (441 nm). We repeated more experiments with fluorescent beads and we used the width of
the Gaussian function at $1/e^2$ of its peak, which is a better estimation of the spot size (now 700 nm). In Fig. 8
(Fig. 7 of the original manuscript), we show the results of in-plane xy raster scans of points with 1 μm steps,
in which the pH value of each point is associated to the relative 1 $\mu\text{m} \times 1 \mu\text{m}$ square of the mesh. This is an
approximation for graphical purposes. In reality, each colored square of the map should be reshaped as a
circle and scaled down to a diameter of 0.7 μm . We added to the caption of Fig. 8 that each point of the maps
corresponds effectively to a spot of 700 nm diameter. We hope that we may be granted of this graphical
approximation. Nonetheless, for the scope of the present investigation, which is the highly localized
measurement of cell outer surface pH, the spatial resolution that matters the most is the one given by the
distance of the sensor from the cell outer surface and the size of the sensor itself. According to our results,
the nanoparticles are clearly attached to the outer surface and the size of the sensor ranges from 90 nm (1
AuNP) to 400 nm (biggest observed nano-aggregates). In other words, the size and the location of the pH-
probe are suitable to investigate the layer of the extracellular milieu in which steep pH gradients manifest
themselves. Please refer to the new text at pg. 6, from Line 16. See also the answer to Comment 2 of
Reviewer #3.

We added new text in the *Conclusion*, at pgs. 13 and 14, in which we mention the interesting method
introduced in ACS Nano DOI: 10.1021/acsnano.8b02200. The pH assessment based on surface energy
transfer using gold nanoparticles and fluorescence dyes reduces the rates of laser-induced photo-bleaching in
multi-hour live-cell experiments. This method can certainly be extended to cell outer membrane surface
analysis, by taking advantage of part of the knowledge introduced in our paper. Nonetheless, the spatial
resolution will be equal to our SERS-based method, as well as the “time window” in which extracellular
surface pH can be reliably measured, which is controlled by the endocytosis of the AuNP.

5. Figure 7c studied the change of average pH value on cell surfaces with and without the presence of EIPA.
To further validate the strategy, an established method detecting the pH variation caused by EIPA should be
used as a control.

We provided standard measurement of cytosolic pH variation induced by EIPA during the acute phase using
pH-sensitive fluorescent dye. Although these pH data are not comparable to the highly localized
measurements of surface pH, the drop of intracellular pH during the acute phase confirm the inhibition of
Na^+/H^+ exchangers and the rise of external surface pH measured by SERS. See the new Fig. S7 in
*Supporting information*; the text at pg. 12, from Line 8 and the experimental procedure added in paragraph
*Confocal fluorescence microscopy* at pg. 16.

6. The TEM images in Figure 4 clearly showed AuNP aggregation and a heterogeneous distribution on cell
surfaces. It is unclear if the local SERS signal difference is due to pH variation on cell surfaces or the
difference of local AuNP density. An experiment needs to be designed to address this question. For example,
fixing the cells after AuNP conjugation on cell surfaces may eliminate local pH difference. Can the SERS
signal become homogeneous?

We performed the requested experiment and the data were added in the new Fig. 8 of the revised version,
which presents the results of 4 cases: MKN-28 with and without addition of EIPA; HepG2 with and without
addition of glutaraldehyde/paraformaldehyde for fixation. Please refer to pg. 12, from Line 12, where new
text was added to introduce the new data of Fig. 8 (d).

Reviewer #2 (Remarks to the Author):

Using 4-MBA functionalized SERS-active substrates (either AuNPs (refs 19-22) or Au nanostructures (e.g.,
Biosensors and Bioelectronics 73 (2015) 202–207) for cell pH sensing have been reported. From this point of
view, this method is not new. The unique of this work lies on the using of NHS-B to un-specifically link to
membrane proteins and then using streptavidin to further link HPDP-B to 4-MBA functionalized AuNPs. In

this way, AuNPs are located at the outside of cell membrane close to ion channels and transporters, enabling
pH sensing locally and even after treated with NHE inhibitor, while AuNPs are located in the intracellular in
the previous reported works (refs. 19-22). However, the work (*Biosensors and Bioelectronics* 73 (2015) 202–
207) reported the direct and accurate mapping extracellular pH of living cells in complex media using SERS
based on 4-MBA functionalized Au nanostructure arrays. Therefore, in general, the work is an increment,
lacks of novelty.

We were aware of that paper, but we believe that the spatial resolution of our method can be compared to
that reported by Anderson et al. (Ref. [12]), which, on the other hand, can be applied only to cancer cells
because based on the use of low-pH insertion peptides and it may be limited by photo-bleaching of the
fluorescent dye. The method proposed in *Biosensors and Bioelectronics* 73 (2015) 202–207 by Sun *et.al.* is
certainly remarkable. It enabled to visualize surface acidosis in HepG2 liver cancer cells, but the mean value
of pH was 7.2, namely much more alkaline than those reported in previous investigations on extracellular pH
of solid hepatocellular carcinomas (see Ref [19-21] and the text added in the introduction at pg. 3 and 4).
Accordingly, if the pH-sensitive probe on the substrate of Sun *et al.* had been indeed entirely attached to the
cell surface, it should have detected higher proton concentration. In the revised paper, we present new data
collected from HepG2 using our method. Cell surface pH was estimated as 6.7, much more acidic than pH
reported by Sun *et. al.*. Please refer to the new Fig. 8. In the manuscript, starting from the last two lines of pg.
12, we address this issue. We explain that HepG2 cells exhibit high concentration of filopodial protrusions
formed on the membrane (see Ref. [55, 56]). The TEM image in Fig. 3 (a) of our revised manuscript clearly
shows the presence of micrometric filopodia also on the basal surface of MKN-28 cells. From this
morphological characterization it follows that in Sun *et. al.* the cells adhered to the gold nanostructured
substrate through filopodia, while an extended part of 4-MBA SAM probed by the laser was “untouched”,
but separated at micrometric distance from the membrane, where the effect of the buffer solution affected the
pH results. In addition, the method reported by Sun *et. al.* can be applied only to study the basal side of the
cells, while in our method the nanoparticles can be attached to both apical and basolateral side of cells,
which may be important for future studies on epithelial cells. The pores of permeable membranes for
epithelial cell culture enable the biotinylation of the basolateral surface and the attachment of nanoparticles
(we are currently making experiments on human bronchial epithelial cells). Based on these considerations,
we believe that the difference between the two methods is obvious, not only in terms of experimental
approach but, by far, in terms of enhanced spatial resolution and localization of the surface pH assessment.
For this reason we did not add the paper of Sun *et al.* to the discussion of the previous manuscript that was
originally conceived as a short communication. We regret about that choice because we now acknowledge
that supplemental experimental evidence and discussion concerning this issue were necessary.

We hope that these new evidence and arguments may convince the Reviewer to reconsider her/his previous
opinion about lack of novelty in our work.

1. The assignment of 1068 and 1588 cm⁻¹ peaks in the sentence on P.6 line 207 is wrong, i.e., “...bands
located at 1068 and 1588 cm⁻¹ are due to ν_{8a} and ν_{12} aromatic ring vibrations, respectively,...” The bands
located at 1068 and 1588 cm⁻¹ should be due to ν_{12} and ν_{8a} aromatic ring vibrations, respectively.
Therefore, the sentence on p. 7 line 229 “... ν_{8a} or ν_{12} aromatic ring vibrations, namely bands C+H or D+I in
Fig. 5 (a),...” is also wrong.

The Reviewer is correct and we thank him/her for noticing our careless mistake. See the corrections at pg. 8,
Line 20 and pg. 10, Line 2.

2. The assignment and explanation about 1390 and 1415 cm⁻¹ bands could also be questionable. In the work
reported in *Biosensors and Bioelectronics* 73 (2015) 202–207, it showed the peak intensity change of two
peaks versus the pH values from 4.05 to 8.99. In that work, Au nanostructure instead of AuNPs was used as
the SERS-active substrates, so there is less or no chance for 4-MBA molecules that have form SAM to
undergo a dramatic orientation change.

In *Biosensors and Bioelectronics* 73 (2015) 202–207 (hereafter referred to Ref. [18], as in the revised
manuscript), the COO⁻ symmetric stretching was studied considering only one band, whose increase of
intensity due to increasing pH was noticed along with the peak shift toward higher frequencies. The observed
shift was attributed to some 4-MBA molecules involved in hydrogen bonding and to the possible interaction
between COO⁻ groups and the ring hydrogens, but the clear asymmetric nature of this band at neutral and
alkaline pH was neglected.

In the revised manuscript, we decided to mention also the explanation of the shift proposed in Ref. [18],
based on hydrogen bonding. Nonetheless, we also keep the characterization of band shift and the formation
of a clear shoulder band according to our original interpretation, which was suggested also by Michiota et al.
[Ref. 33]. We added Ref. [35], in which calculation based on density functional theory on the adsorption
behavior of 4-MBA on silver colloids demonstrated that, depending on pH, the protonated molecules are
bonded to the metal surface only through the S atoms, while, in the deprotonated form, through the S atoms
and the COO⁻ groups with a tilted orientation, causing significant intensity shift and the onset of a shoulder
band. Moreover, we added Ref. [36], which demonstrated that benzoic acid, in which thiol group is not
present, conjugates to gold substrates via carboxylate group and the COO⁻ symmetric stretching band is
centered at around 1370 cm⁻¹, namely gold surface can interact with COO⁻, inducing massive shift of the
Raman stretching band to lower frequencies. Please refer to the new text added from the first line of pg. 8.
Independently on which of the two explanations is more plausible, we believe that the deconvolution of the
Raman intensity around 1400 cm⁻¹ with 2 bands is the most correct approach. In fact, this approach considers
the contribution of every COO⁻ group forming the SAM, namely it is more accurate. Also in Fig. 3 (a) of Ref.
[18] it is clear that, at alkaline pH, the band peak shifts to higher frequencies, but also a shoulder at lower
frequency is formed. The frequencies of the shoulder sub-band are the same frequencies of the band at more
acidic pH, namely the sub-band is still associated to -COO⁻ stretching. Nonetheless, this shoulder was
neglected in pH calculation of previous studies.

3. Caption of Figure 5a should provide the ratio of 4-MBA/HPDP-B. The 2-pyridine shown in Figs. 5a and
5b could cause a confusion as there is no 2-pyridine was immobilized on AuNPs. Why are the four strong
peaks shown in Fig. 5b due to HPDP-B related to 2-pyridine? Due to the smaller Raman scattering cross-
section of HPDP-B compared to 4-MBA, it is reasonable that the intensity is about one order of magnitude
lower than 4-MBA in Fig. 5a. The four strong peaks in Fig. 5b should be more related to O=C-NH as they
are more close to sensing surface.

We added the ratio of 4-MBA/HPDP-B in the new Fig. 5 (a) and in its caption.

In addition, we reported a new spectrum of HPDP-B in Fig. 5 (b), from AuNP colloidal solution conjugated
with higher concentration of HPDP-B, which was equal to the concentration used for 4-MBA in Fig. 5(a).

In the previous version we showed the spectrum at low concentration, which is the HPDP-B concentration
used for conjugation of the pH-sensor. The aim of the previous version was the confirmation of HPDP-B
conjugation at low concentration and its negligible Raman signal as compared to 4-MBA SAM signal. In the
new version, the HPDP-B spectrum is more intense and the detailed characterization of the SAM is possible.
In Fig. S3 of *Supporting information*, we show the HPDP-B spectrum compared to the SERS spectrum of 2-
pyridine thiolate. The assignments are reported in Table S1 of *supporting information*.

The HPDP-B molecule is composed of a pyridyldithiol group linked to a biotin residue via a 2.9 nm hexyl
arm spacer. The disulfide bridge reacts with the gold surface, leading to dissociative adsorption of HPDP-B
through gold-atom-mediated bonds as 2-pyridine (2-Py) thiolate and biotin-hexyl spacer arm thiolate.

We did not detect the S-S stretching band at around 500-515 cm⁻¹, namely the disulfide bond was broken as
expected (i.e., we added also new references [37] and [38]). The most intense bands of 2-Py thiolate are also
observed in the HPDP-B spectrum, since this molecule is close to the sensing surface. In the HPDP-B
spectrum, we also observed other less intense bands that were associated to the biotin-hexyl spacer arm
thiolate. Please refer to the new text added from Line 23 of pg. 8.

4. As reported in *Biosensors and Bioelectronics* 73 (2015) 202–207, other cations, especially divalent cations
(Ca²⁺, Mg²⁺) can cause the intensity change and the peak shift of ~1420 cm⁻¹ peak. In this study, as stated
on p.11 lines 358-361, “During the spectroscopic analyses we used an isotonic buffer solution containing 115
mM NaCl, 15 mM NaNO₃, 5 mM KCl, 1 mM Ca(NO₃)₂, 1 mM Mg(NO₃)₂, 10 mM HEPES, 10 mM
glucose, 10 mM sucrose and adjusted to pH 7.4 by adding CsOH.” Therefore, the calibration curve (Fig. 6a)
should also be done in such solution. While this won’t affect the trend by adding EIPA, the real pH values
could be different.

The reviewer is absolutely right and we thank her/him. We repeated the calibration with AuNP in isotonic
buffer solutions at different pH. The new calibration is shown in the new Fig. 7(a). The divalent cations
clearly affected the band ratio, similarly to *Biosensors and Bioelectronics* 73 (2015) 202–207 (Ref. [18]).

The comparison between old and new calibration (PBS vs. isotonic buffer solution used for cells) is reported
here only for the Reviewers, since it is not pertinent to final scope of the paper.

Accordingly, we recalculated the pH values on the surface of the analyzed cells. The results are shown in the
new Fig. 8. Please note that the cell surface pH values in MKN28 with and without EIPA are now about 0.2
lower than those reported in the original manuscript. The new results are certainly more accurate and they
can finally be compared to those obtained in Ref [12] in the case of cancer cells in the presence of glucose,
as in our experiments. In fact, in our results based on the previous calibration, the MKN28 surface was more
acidic than the cancer cells of Ref. [12] analyzed without glucose, but more alkaline than the cancer cells of
Ref. [12] with glucose. Since in our buffer solution we had glucose, we are happy that our corrected results
are more consistent with the previous results of Ref. [12]. Please refer to the new Fig. 8 and the correction
from Line 17 of pg. 13.

5. The 2D pH mapping via the xy raster scan shown in Figure 7 should provide the z position. As shown in
Figure 2 of Supporting Information, SERS spectra signal only appeared locally where AuNP aggregations
are. Also, for Figure 2 of Supporting Information, it also needs to indicate where this z-scan was done. My
guess is that this should not be in the middle of the cell as the strong SERS signal from the middle of the z-
scan range. This z-scan might be taken close to the edge of the cell.

Please refer to the new explanation added from Line 4 from the bottom of pg. 10. The z-coordinate of each x-
y raster scan was set after estimation of the average AuNP position in the middle of each analyzed cell by z-
axis line scans. Considering the micrometric axial length of the laser probe, this procedure enabled to
illuminate the AuNP attached to the surface in the majority of the investigated spots. The axial length of the
laser probe was showed in the new Fig. S5 in Supporting information.

The scan in Fig. 2 of Supporting information in the previous version is now modified for clarity sake and
reported in the new Fig. S4 of Supporting information. We apologize for the lack of clarity in the explanation
that may have confused the Reviewer. The scan was performed in the middle of the cell. It is a z-scan, from
the bottom glass upon which the cell was seeded to 20 μm above the bottom glass, across the cell. The
position of the maximum intensity at $z=12 \mu\text{m}$ is the location of the AuNP, namely the cell membrane, since
we demonstrated that AuNP are attached to the outer cell surface. In the new Fig. S4, we added a sketch of
the cell cross-section along which the z-axis line scan of Raman spectra was collected. We added in the
caption of Fig. S4 that the z-axis line scan was performed in the middle of the cell.

6. The claim of two peaks at 999 and 1021 cm^{-1} , which are associated with the decomposed aromatic species
(i.e., TP), needs to be careful. Why there are no obvious such two peaks in the pH calibration experiments,
i.e., Figure 6, for both acidic ($\text{pH} = 6.0$) and basic ($\text{pH} = 8.0$), compared to the detection of cells (Fig. 7)?
The increase of these two peaks could be due to the orientation change of 4-MBA. The increase of these two
peaks upon the binding of fructose on 4-mercaptobroic acid SAM was reported in ACS Nano 9, 2668 (2015).

In Fig. 6 of the original manuscript, we purposely selected two spectra at different pH but similar lack of
transformation. The aim of the figure is to emphasize the variation of the intensities A+B and E as a function
of pH. For this purpose, we found appropriate to choose two spectra with similar but marginal onset of F, G,
H, and I bands. Please consider that, at laser power of 20 mW, in colloidal solutions we did not detect

spectral transformation in every measurement. In the new manuscript we added new arguments and new
experiments to validate our theory of SAM transformation. Please refer to Fig. 6 and the discussion from
Line 12 of pg. 9. The power of the laser indisputably effected the transformation, which was not observed in
every measurement obtained with 1mW (note that in Ref. [18], for example, transformation was not
observed because they used 1 mW). At maximum power (i.e., 72 mW) we always observed the onset of F, G,
H, and I bands. The positions of the new bands, as well as the ratios of their intensities, clearly resemble the
spectrum of TP shown in Fig. 5 (c). Nonetheless, we mentioned also the interpretation of Sun *et. al.* (Ref.
[46]). As suggested by the Reviewer, they hypothesized that the origin of these bands in 4-
mercaptophenylboronic acid (4MPBA) affected by fructose may be due to reorientation and charge
redistribution of the benzene ring. Conversely, Ly et al. (doi.org/10.1002/sia.6184) observed deboronation
reaction of 4MPBA via fructose and glucose on silver surfaces by SERS at the excitation wavelengths of 488,
514, and 633 nm. They found all the characteristic bands of TP as we did. The new Fig. 6 (h) demonstrates
that, although we can debate the origin of the new bands, when they are observed, the deconvolution in sub-
bands and the use of band C is essential for precise pH calculation.

7. In the Figure 3 of Supporting Information, the optical images show clearly that there are many AuNPs at
the edges of cells. Did authors collect the SERS data from those areas? How about the pH values of those
areas?

Please refer to the new text at pg. 11, from Line 18. We also prepared a new Fig. S6 in *Supporting*
*information*. The average pH measured from spectra of micro-aggregations is 7.41. An example of spectra
collected from micro-aggregation is showed in Fig. S6 (a). In Fig. S6 (b), we compared the intensities of
Band C collected from micro-aggregations, colloidal solutions and x-y raster scans on the surface of cells
without micro-aggregations. There is a clear difference between the intensities detected on micro-
aggregations and the investigated cell surfaces.

8. CFM was applied to confirm the label was only at the membrane of cells. However, as shown in TEM and
even optical images, the AuNPs can be clearly seen. Therefore, CFM method may not be needed.

We moved Fig. 2 of the previous version to the *Supporting information* (Fig. S1). We would like to preserve
Fig. 3 of the previous version, which is now Fig. 2 in the revised version. The TEM images confirm the
AuNP attachment on the outer membrane, but Fig. 3 is useful to prove the distribution of the AuNP on the
cell surface (*xy plane*).

9. Figure 1 of Supporting Information needs to provide the z values for each line scan.

We added the information as requested. Figure 1 of Supporting Information is Fig. S2 in the *Supporting*
*information* of the revised manuscript.

10. The wavenumber of peak around 1600-1670 cm⁻¹ was labeled differently in the Figure 7 caption (1063
361 cm⁻¹) but 1068 cm⁻¹ in the legend.

Thank you for noticing this careless mistake. Figure 7 is now Fig. 8 and the caption was modified to explain
the new implemented figure and to fix the mistake (i.e., 1068 instead of 1063).

Reviewer #3 (Remarks to the Author):

1. For the calibration curve, the SERS active modes coming from HPDP B and TP molecules seems to be
taken into account. However in the real samples also the Raman bands of Streptavidin should experience
significant SERS enhancement. Have the authors considered this issue?

According to our protocol, streptavidin (SA) first reacts with biotinylated surface proteins; then, only after
removal of unreacted SA, we add the functionalized AuNP to the cell sample. The AuNP were separately
conjugated to 4-MBA and HPDP-B. This is a crucial feature of the experimental method, since it guarantees
that each AuNP attached to the cell membrane has approximately only one SA molecule close to its surface,
namely the SERS signal of SA does not affect the overriding signal of the pH-sensitive 4-MBA self-
assembled monolayer (SAM). In addition, the Raman scattering is not enhanced in every location of the gold

surface, but only on the surface of some hot spots. The hot spots are concentrated on the tips of the nano-
urchins and in the gap between two adjacent AuNP in a nano-cluster. Moreover, the enhancement decreases
exponentially from the surface (i.e., max) to about 10-15 nm away from the surface (i.e., negligible
enhancement). The small 4-MBA is adsorbed on the surface and the SAM is only 0.78 nm thick (Ref. [30]).
Conversely, the SA attached to AuNP is some nanometer away from the surface, considering the HPDP-B
arm spacer length of 3.9 nm. In summary, i) the ratio between 4-MBA and SA on the gold surface is
massively in favor of 4-MBA; ii) the probability that SA is located in the hot spot is also small; iii) the
enhancement factor on the SA attached to gold is smaller than the enhancement on the 4-MBA molecules
stuck on the surface. Please refer to the new text at pg. 4, Line 6 from the bottom.

2. The resolution of the average pH measurements are diffraction limited to the size of the laser spot,
therefore the SERS signals are the average within this area and also the pH values. It is not clear why the
axial resolution is diffraction limited while the resolution along the other directions is dictated by the size of
the Au NP degree of agglomeration. This point should be explained more clearly.

We added new explanation of the spatial resolution at pg. 6, from Line 16. In the new Fig. 3 (b) we also
added an explanatory description of the SERS pH-probe given by the interaction between the laser beam and
the AuNP. The in-plane spatial resolution of the measurement can be approximated as the size of the laser
spot. Conversely, we define the axial resolution as the maximum distance from the cell surface at which the
pH is measured. Since the SERS signal is merely originated from the 4-MBA SAM (i.e., 0.78 nm thickness,
depicted with green lines in Fig. 3(b)) the axial resolution can be approximated as the size of the nano-
sensors. For example, if we consider one approximately spherical AuNP attached to the surface, it can
probe pH as close as in the point of attachment and as far as in the diametrically opposite point of the AuNP.
The pH calculated by SERS is approximately the average. See the following sketch:

3. Can the author estimate the SERS enhancement factor of the relevant molecular probe involved (MBA)
and its dependence on the degree of Au NP agglomeration. This issue is very important since the value of pH
depends on the SERS intensity of the relevant bands and their intensities should also depend on the degree of
agglomeration and the average distance between Au NPs.

We thank the reviewer for the suggestion. We calculated the enhancement factor. See the new text at the end
of the paragraph *Measurement of pH using AuNP and SERS* (pg. 10); the details of the calculation in the
paragraph *Enhancement factor calculation* of the section *Methods* (pgs. 17 and 18) and the new Fig. S9.

ADDITIONAL INFORMATION FROM THE AUTHORS:

Figure S8 of *Supporting information* is the modified version of the previous Fig. 3 of *Supporting information*.
We show now 4 additional examples of hyperspectral maps of surface pH from MKN28, HepG2 and
MKN28 treated with EIPA.

Reviewers' comments:

Reviewer #1 (Remarks to the Author):

The authors have addressed all my questions. The quality of this manuscript has been significantly improved. However, one concern of this manuscript is still the resolution limit, which is not better than existing pH detection methods.

Reviewer #2 (Remarks to the Author):

In the revision, authors carried out more experiments and revised the manuscript carefully to address the reviewer's comments. While the quality of the work has been significantly improved, some discussions need to be careful to avoid misleading.

1. Regarding the answer to the second question, the reviewer would suggest the authors pay attention to the following points. (a) The condition that the 4-MBA SAM is formed. In both refs 33 and 35, the 4-MBA SAMs were formed in AQUEOUS solution under different pH values. No doubt, under alkaline condition in aqueous solution, COOH groups are deprotonated, so FREE 4-MBA molecules can adsorb on Au (Ag) surface via thiol to form S-Au(Ag) bond and via COO- to form COO-Au(Ag) bond. As pointed in ref. 35, under acidic and neutral conditions (in aqueous solution), 4-MBA adsorb on Ag via strong Ag-S bond, while weak COO-Ag and S-Ag in alkaline condition. In this work, the 4-MBA SAM was formed in PBS buffer condition according to the experimental description. So, majority of 4-MBA (if not all) should adsorb on Au surface via strong Au-S bond. In ref. 18, however, the 4-MBA SAM was formed in ethanol solution, so the binding should be Au-S. (b) Once the 4-MBA SAM has been formed via the Au-S bond, all the 4-MBA molecules have been anchored on the Au surface. In this work, after forming 4-MBA and HPDP-B SAM, "the Au NPs were collected after centrifugation and resuspended in PBS at pH 7.4 for cells treatment or in isotonic buffer solution at different pH for the calibration of the nanosensor (i.e., in the pH range 4 to 10)." So, the COOH group (unbounded) will be protonated or deprotonated in response to the solution / environment pH change. Two possible situations could cause COO-Au interaction even though SAM has been formed: (i) two gold nanoparticles are very close (could be the case in this work since TEM images show Au NP clusters, but this should not be the case when authors did pH calibration in bulk solution), but this is still rare given each NP already has a SAM layer; (ii) multilayer 4-MBA formed in the SAM forming procedure as pointed in ref. 35 (and other people also reported that multilayer instead of monolayer could be formed). In this case, the outer layer 4-MBA molecules that are deprotonated in alkaline condition could diffuse to Au surface and bind via COO-Au bond. Taking these two points into account, the sentences starting with "On the other hand..." in redline on p. 8 lines 256-264 could be misleading. In short, when 4-MBA molecules are free in aqueous solution, under alkaline condition, they can form S-Au and COO-Au bond, but once the SAM has been formed via Au-S, these already bounded molecules have no option to form COO-Au bond even under alkaline condition. The fact of asymmetric peak around 1400 cm⁻¹ was definitely observed in many publications (like ref. 22 as well). It for sure needs more investigation. Nonetheless, the reviewer feels the discussion on p. 8 lines 256-264 could be misleading.

2. Based on authors' assignment of 1370 cm⁻¹ peak due to COO-Au, then these COO are not free, they can't respond to pH change, then they should not be counted in the intensity ratio (I_{A+B}/I_C) for the calibration curve. Did authors try to plot the calibration curve using I_B/I_C?

Reviewer #3 (Remarks to the Author):

In this revised version of the manuscript the authors have performed significant improvements clarifying in a satisfactory way all the main issues pointed out by myself. In this respect they have clarified the important feature concerning the spatial resolution of the pH sensor and have also analysed the relevant

reporter molecules that are used for pH estimation as well as performed an estimation of the SERS enhancement factor (lower bound).

Therefore, I consider that the main concerns have adequately been answered and the manuscript is now suitable for publication in Nature Communications

Response to Reviewers' remarks:

We truly appreciated the Reviewers' comments, opinions and support to prepare a revised version of our manuscript. We would like to thank the 3 Reviewers for the valuable time they have spent reviewing our work. The revised parts in the new manuscript and the answers to each comment in this document are written in red:

Reviewer #1 (Remarks to the Author):

The authors have addressed all my questions. The quality of this manuscript has been significantly improved. However, one concern of this manuscript is still the resolution limit, which is not better than existing pH detection methods.

In the first revision of the manuscript we addressed the issue of the spatial resolution. In the previously revised manuscript, we acknowledged that pH assessment based on surface energy transfer using gold nanoparticles and fluorescence dyes has similar spatial resolution of our method. Nonetheless, the use of this technique for the measurement of extracellular pH on the outer surface of the cells has not yet been exploited and published. In addition, the use of this technique should take advantage of part of the knowledge presented in our paper to effectively attach the pH-sensor to the outer membrane. We amply discussed in the manuscript the method based on low-pH insertion peptides, which has been thoroughly compared to our method. In conclusion, as compared to other techniques that have similar spatial resolution, we would like to remark that our experimental procedure can be implemented in a simple and nimble way, since the cell preparation before Raman analysis requires only a couple of hours. The method was purposely developed using commercially available materials, which are easy to find. In addition to ex-vivo analysis, it has the potential to combine in-vivo cell sorting for diagnostic purpose and phototherapy: in fact, a calibrated increase of power output in the micrometric laser focus induces cell death due to the highly localized heat generated on the gold metal surface. At the completion of the surgical procedure, the nanoparticles attached to the healthy tissue may be washed away, since they are attached to the external membrane without endocytosis and the bonds are cleavable. Obviously, this future potential application of the technique has to be carefully evaluated, developed and tested to prove to be effective. Nowadays, the use of advanced spectrometer for tissue imaging combined to data discrimination by machine learning-based cell classifiers is gaining more and more popularity. Hopefully, such interest will foster the fast improvement and refinement of the technique. In light of these considerations, we hope that the manuscript may be worthy of publication in Nature Communications.

Reviewer #2 (Remarks to the Author):

In the revision, authors carried out more experiments and revised the manuscript carefully to address the reviewer's comments. While the quality of the work has been significantly improved, some discussions need to be careful to avoid misleading.

1. Regarding the answer to the second question, the reviewer would suggest the authors pay attention to the following points. (a) The condition that the 4-MBA SAM is formed. In both refs 33 and 35, the 4-MBA SAMs were formed in AQUEOUS solution under different pH values. No doubt, under alkaline condition in aqueous solution, COOH groups are deprotonated, so FREE 4-MBA molecules can adsorb on Au (Ag) surface via thiol to form S-Au(Ag) bond and via COO- to form COO-Au(Ag) bond. As pointed in ref. 35, under acidic and neutral conditions (in aqueous solution), 4-MBA adsorb on Ag via strong Ag-S bond, while weak COO-Ag and S-Ag in alkaline condition. In this work, the 4-MBA SAM was formed in PBS buffer condition according to the experimental description. So, majority of 4-MBA (if not all) should adsorb on Au surface via strong Au-S bond. In ref. 18, however, the 4-MBA SAM was formed in ethanol solution, so the binding should be Au-S. (b) Once the 4-MBA SAM has been formed via the Au-S bond, all the 4-MBA

molecules have been anchored on the Au surface. In this work, after forming 4-MBA and HPDP-B SAM, “the Au NPs were collected after centrifugation and resuspended in PBS at pH 7.4 for cells treatment or in isotonic buffer solution at different pH for the calibration of the nanosensor (i.e., in the pH range 4 to 10).” So, the COOH group (unbounded) will be protonated or deprotonated in response to the solution / environment pH change. Two possible situations could cause COO-Au interaction even though SAM has been formed: (i) two gold nanoparticles are very close (could be the case in this work since TEM images show Au NP clusters, but this should not be the case when authors did pH calibration in bulk solution), but this is still rare given each NP already has a SAM layer; (ii) multilayer 4-MBA formed in the SAM forming procedure as pointed in ref. 35 (and other people also reported that multilayer instead of monolayer could be formed). In this case, the outer layer 4-MBA molecules that are deprotonated in alkaline condition could diffuse to Au surface and bind via COO-Au bond. Taking these two points into account, the sentences starting with “On the other hand...” in redline on p. 8 lines 256-264 could be misleading. In short, when 4-MBA molecules are free in aqueous solution, under alkaline condition, they can form S-Au and COO-Au bond, but once the SAM has been formed via Au-S, these already bounded molecules have no option to form COO-Au bond even under alkaline condition. The fact of asymmetric peak around 1400 cm^{-1} was definitely observed in many publications (like ref. 22 as well). It for sure needs more investigation. Nonetheless, the reviewer feels the discussion on p. 8 lines 256-264 could be misleading.

The comment and suggestion from the Reviewer are correct and pertinent. We agree that, once the SAM is formed by thiol adsorption, the variation of pH did not lead to “molecular flip-flop” associated to COO⁻ adsorption on the gold surface. In the first revised manuscript, we hypothesized that the SERS intensity of band A at around 1390 cm^{-1} may be due to S-Au bonded deprotonated molecules in which the carboxylate groups interact with the metal surface. References 33, 35 and 36 of the previously revised manuscript support the intrinsic affinity between carboxylate groups and Au. In case of SAM formed at low pH via COOH adsorption, the symmetric stretching band was observed at 1370 cm^{-1} (Ref. 36), while in our case the position of band A is at around 1390 cm^{-1} , namely an intermediate position between adsorbed carboxylates and band B (i.e., free carboxylates of the more vertically oriented 4-MBA molecule). This is the concept that we wanted to articulate at pg. 8, when we stated that “These evidences strongly suggest that in the case of 4-MBA, the SERS intensity at around 1400 cm^{-1} is due to carboxylates groups differently bound to the metal surface, depending on pH”. In our interpretation, the intensity around 1400 cm^{-1} is the convolution of bands A and B: both come from S-Au bonded deprotonated 4-MBA, but in band A the molecules are tilted and the COO⁻ group may be affected by the gold surface. The following figure shows the SERS intensities collected from AuNP at pH 4.0, 6.4 and 7.4 in the range $625\text{--}785\text{ cm}^{-1}$. The intensities were normalized using band C located at 1068 cm^{-1} (ν_{12} ring vibration).

The bands at 694 and 720 cm^{-1} are assigned to out-of-plane $\gamma(\text{CCC})$ vibrations in acidic and neutral/alkaline environments, respectively (Phys Chem Chem Phys., 2015;17(39):26093-100. doi: 10.1039/c5cp03844h). From these data, as shown in the explanatory sketches of the figure, we inferred that some 4-MBA molecules of the SAM are tilted toward the gold surface (as also observed in other papers). The reduced distance between the gold surface and the carboxylates of the tilted 4-MBA molecules may explain the interaction causing stretching of the C-O bonds, namely redshift of the Raman frequencies associated to COO^- symmetric stretching. Nonetheless, we agree that the conclusive clarification of the origin of band shift and asymmetry requires more investigation. The explanation of the COO^- stretching band shift and asymmetry reported in previous studies is based on hydrogen bonding and the possible interaction between COO^- groups and the ring hydrogens (Ref. 18, 22 and the new Ref. 35 of the second revised manuscript). This explanation makes sense and it also support the choice of the fitting procedure based on 2 bands, which is the focal point in our paper. Both bands are pH responsive. At acidic pH only Band A is observed and its intensity disappears at highly acidic pH. The latter evidence confirms that in the conjugated AuNP every 4-MBA molecule is pH responsive and there is no adsorption of COO^- groups. The separation between band A and B becomes evident as pH rises to the alkaline range. Based on these considerations, we changed the text at the end of pg. 7: we removed the hypothesis of carboxylates-gold interaction and we reported only the intramonolayer hydrogen bonding between adjacent 4-MBA molecules. We removed the previous references 35 and 36. We added a new reference (Ref. 35 in the latest manuscript, Bishnoi, S. W. et al. All-optical nanoscale pH meter. Nano letters 6, 1687-1692 (2006)), which also reports asymmetry and shift of the band due to hydrogen bonding. Again, we thank the Reviewer for noticing this point in our discussion that may have been misleading.

Please note that the sentence “This reaction is favored at alkaline pH (i.e., $\text{pH}>9$ in our experiments), where the deprotonated 4-MBA is predominant and, consequently, the population of non-bonded $-\text{COO}^-$ groups increases along with the probability of plasmon-induced catalytic reaction forming CO_2 and TP” at pg. 9 of the first revised manuscript was replaced by “This reaction is favored at alkaline pH (i.e., $\text{pH}>9$ in our experiments), where the population of deprotonated 4-MBA increases along with the probability of plasmon-induced catalytic reaction forming CO_2 and TP” at the end of pg. 8 of the second revised manuscript.

Moreover, in line 6 of the paragraph *Outer cell membrane surface labelling* we changed “conjugate via carboxylate- and thiol-gold interactions” to “conjugate via thiol-gold interaction”. In fact, the combination of “conjugate”, “carboxylate” and “interactions” is misleading because the reader may infer that, in our SAM, COO^- are adsorbed on the surface.

2. Based on authors’ assignment of 1370 cm^{-1} peak due to $\text{COO}-\text{Au}$, then these COO are not free, they can’t respond to pH change, then they should not be counted in the intensity ratio ($I_{\text{A+B}}/I_{\text{C}}$) for the calibration curve. Did authors try to plot the calibration curve using $I_{\text{B}}/I_{\text{C}}$?

As stated in the answer to Comment 1, the intensity of band A responds to the variation of pH and it is undetectable at highly acidic pH. It implies that every COO^- groups associated to this band are not adsorbed on the gold surface and they can be protonated. The plot of $I_{\text{B}}/I_{\text{C}}$ is as follows:

The data still show the typical trend of Henderson–Hasselbalch equation. We can notice a discontinuity at pH approaching 7.0. In fact, at more acidic pH the SERS intensity of the COO^- symmetric stretching is prevalently represented by band A.

Reviewer #3 (Remarks to the Author):

In this revised version of the manuscript the authors have performed significant improvements clarifying in a satisfactory way all the main issues pointed out by myself. In this respect they have clarified the important feature concerning the spatial resolution of the pH sensor and have also analysed the relevant reporter molecules that are used for pH estimation as well as performed an estimation of the SERS enhancement factor (lower bound). Therefore, I consider that the main concerns have adequately been answered and the manuscript is now suitable for publication in Nature Communications.

We thank the Reviewer for the positive comments and for her/his contribution throughout the entire revision process.

ADDITIONAL INFORMATION FROM THE AUTHORS:

At pg. 18, paragraph *Statistical analysis*, we changed “non-paired t-test” to “two-tailed unpaired t-test”.

REVIEWERS' COMMENTS:

Reviewer #2 (Remarks to the Author):

In this second revision, authors have addressed my questions and comments in the first round review. I would recommend for publication in Nature Communications.